# Mapping stratospheric nitric acid (HNO₃) patterns in the extratropics with nadir-viewing infrared sounders – a retrieval perspective

Nadia Smith[1*], Michelle L. Santee[2], Christopher D. Barnet[1*]

[1]Science and Technology Corporation, Columbia, Maryland, USA
[2]Jet Propulsion Laboratory, California Institute of Technology, Pasadena, California, USA
[*] Retired

*Correspondence to*: Nadia Smith (nadia.smith.work@gmail.com)

**Abstract.**

With this paper, we aim to demonstrate how stratospheric HNO₃ can be retrieved from nadir hyperspectral infrared (IR) measurements such that it is largely uncorrelated with tropospheric HNO₃ and most other interfering signals. This is achieved by decomposing the set of HNO₃ sensitive channels into orthogonal vectors that isolate the stratospheric HNO₃ signal for use in the retrieval. Such a nadir-IR HNO₃ product could add useful information to the monitoring of some stratospheric chemical processes affecting ozone in the extratropics, especially once the state-of-the-art Microwave Limb Sounder (MLS) on Aura is decommissioned in 2026. Nitric acid is typically used as indicator species for heterogeneous chemical processing inside the winter polar stratospheric vortices. The proposed stand-alone stratospheric HNO₃ retrieval would be an improvement over the only other nadir-IR HNO₃ product available today, namely FORLI (Fast Optimal Retrieval on Layers for IASI), which is a stratosphere + troposphere correlated profile retrieval affected by uncertainty in tropospheric water vapor at the time of measurement. We demonstrate the potential of this new stratospheric nadir-IR HNO₃ retrieval strategy using the Community Long-term Infrared Microwave Combined Atmospheric Processing System (CLIMCAPS) as the retrieval framework with measurements from CrIS (Cross-track Infrared Sounder) on the Joint Polar Satellite System 1 (JPSS-1) during the Northern Hemisphere winter of 2019/2020. Future work will focus on optimizing and validating CLIMCAPS HNO₃ retrievals for operational deployment.

## 1 Introduction

Measurements of HNO₃ help explain O₃ chemical processes, especially in the extratropical stratosphere, where seasonal O₃ holes form to the detriment of life on Earth. Instruments on Earth-orbiting satellites provide the bulk of the observations necessary to monitor O₃ loss. Two types of modern-era space-based instruments have the ability to observe stratospheric gases, namely limb-viewing sounders, such as MLS (Microwave Limb Sounder; Waters et al., 2006), and nadir-viewing sounders, such as IASI (Infrared Atmospheric Sounding Interferometer; Chalon et al., 2017). The most basic distinction one can draw as far as stratospheric observations go is as follows: limb sounders have higher vertical resolution but more limited spatial

coverage with narrow instrument swath widths, while nadir sounders have lower vertical resolution yet more extensive spatial coverage with much wider swath widths. Compared to MLS, the limited vertical sensitivity of nadir infrared (IR) sounders to HNO$_3$ prohibits the observation of multiple stratospheric layers, which can impede their ability to observe smaller, less predictable events. MLS was launched on the National Aeronautics and Space Administration (NASA) Earth Orbiting System (EOS) Aura satellite in 2004. Two decades later, MLS remains unmatched in its ability to measure the chemical state of the atmosphere, from the upper troposphere to the mesosphere. In fact, MLS observations of atmospheric chemistry (15 species in total) have transformed our understanding of stratospheric processes. In contrast, IASI instruments – like all nadir sounders – measure the atmosphere from the boundary layer to the top of atmosphere (TOA) in wide swaths (~2000 km) from pole to pole. But their orbital configuration is not the only distinction to highlight. IASI is an IR sounder and, unlike microwave sounders such as MLS, cannot measure the atmospheric state through clouds. However, despite its scientific significance, MLS is the only instrument of its kind in operational orbit, and its record of stratospheric observations will end once Aura is decommissioned within the next couple of years. IASI, on the other hand, is one of many IR sounders in space and well supported with plans to continue measuring the atmosphere for the next two decades. A detailed comparison is beyond the scope of this paper, but such a basic distinction serves the purpose of our work described here.

While MLS retrievals of HNO$_3$ have been critically important to understanding and modelling stratospheric processes (e.g., Brakebusch et al., 2013; Santee et al., 2008), the use of IASI HNO$_3$ retrievals has not been as widespread. For those studies that do exist, the IASI HNO$_3$ product is presented as a column-integrated quantity (stratosphere+tropospere) in demonstrations that largely avoid the Arctic region, where vortices are both smaller and shorter-lived than those in the Antarctic (Ronsmans et al., 2018; Wespes et al., 2022). This paper presents a method for HNO$_3$ retrievals from the Cross-track Infrared Sounder (CrIS; Glumb et al., 2002; Strow et al., 2013) that allows the separation of stratospheric HNO$_3$ from the spectrally correlated information content about the troposphere.

As stated earlier, IASI is one of many nadir-IR sounders in space. The Atmospheric InfraRed Sounder (AIRS) initiated the modern era of IR sounding capability when it was launched in 2002 on Aqua (Aumann et al., 2003; Pagano et al., 2003) in a sun-synchronous orbit with ~1:30 am/pm equatorial crossing time. In 2011, CrIS was launched on the Suomi National Polar Partnership (SNPP) payload to continue the AIRS record with observations of the atmosphere at ~1:30 am/pm equatorial crossing time. CrIS has since been launched on a series of payloads, collectively known as the Joint Polar Satellite System series (JPSS+), with JPSS-1 in 2017, JPSS-2 in 2022, and two additional JPSS payloads planned in the next decade. Like IASI, CrIS is poised to continue its record well into the 2040s. Despite this long record, however, a science-quality HNO$_3$ product was largely absent for AIRS and CrIS until CLIMCAPS V2.1 (Community Long-term Infrared Microwave Combined Atmospheric Processing System) was released in 2018 (Smith and Barnet, 2023a). HNO$_3$ was never before considered a target variable in the suite retrieved from the heritage AIRS Level 2 retrieval system (Susskind et al., 2003), and it features in the NOAA-Unique Combined Atmospheric Product System (NUCAPS) suite only as an experimental by-product (Barnet et al., 2021). The full suite of CLIMCAPS V2.1 retrieval products includes atmospheric temperature ($T_{air}$), eight gaseous species

(H2Ovap, CO2, O3, N2O, CH4, HNO3, CO and SO2), as well as cloud and Earth surface properties (see Table 1 in Smith and Barnet, 2023a). The Fast Optimal Retrieval on Layers for IASI (FORLI; Ronsmans et al., 2016), in contrast, retrieves only CO, O3 and HNO3. Analysis of the FORLI HNO3 product for IASI, therefore, relies on estimates of stratospheric $T_{air}$ from external sources, while the CLIMCAPS product from AIRS and CrIS provides coincident observations of $T_{air}$ and HNO3.

The goal of this paper is to demonstrate how stand-alone estimates of stratospheric HNO3 can be retrieved from nadir IR measurements such that they are largely uncorrelated with coincident tropospheric signal and noise (or the signal-to-noise ratio, SNR, in short). This is achieved by decomposing the set of HNO3 sensitive channels into orthogonal vectors that isolate the stratospheric HNO3 SNR from most of the tropospheric SNR that would otherwise correlate with the stratospheric HNO3 retrieval. The work presented here is novel and promises to improve upon the status quo by enabling a stand-alone stratospheric

HNO3 product from nadir IR measurements. We use CLIMCAPS as the bedrock system for this demonstration because it allows the selection of individual eigenfunctions generated by the orthogonal decomposition of the measurement SNR matrix at runtime. As described in detail elsewhere (Smith and Barnet, 2019, 2020), CLIMCAPS dynamically regularizes a subset of orthogonal vectors during its Bayesian inversion to harness the measurement signal when it is high and damp the measurement signal when it is low. FORLI, on the other hand, employs a more traditional Optimal Estimation (OE) approach as put forth

by Rodgers (2000) wherein the IR measurement is regularized using a statistical estimate of uncertainty about the target variable. The FORLI approach to regularization has the disadvantage that it does not account for variation in SNR from scene to scene, which can lead to an over- or under-estimation under some conditions. We show here how the stratospheric HNO3 signal measured by nadir IR sounders, such as AIRS, IASI and CrIS, projects into a single eigenfunction that can be isolated from most of the tropospheric SNR otherwise coincident in the HNO3-sensitive IR spectral channels. Future work will build

on this with an optimization and validation of the CLIMCAPS HNO3 product for operational deployment.

In support of the stated goal, we identified two objectives: (1) determine the CLIMCAPS algorithm configuration that would enable an independent stratospheric HNO3 retrieval and (2) visually contrast our experimental CLIMCAPS HNO3 retrieval against state-of-the-art MLS HNO3 during the Northern Hemisphere winter of 2019/2020 when a strong vortex formed. The MLS algorithm retrieves stratospheric HNO3 and thus allows an apples-to-apples visual comparison with the nadir-IR

stratospheric HNO3 product proposed here. A comparison against FORLI HNO3, which is a stratosphere+troposphere correlated profile retrieval (Ronsmans et al., 2018), would require more sophisticated methods to account for the product differences. Of importance here is demonstrating the viability of the CLIMCAPS retrieval approach for a nadir IR stratospheric HNO3 product.

It should be noted, however, that nadir IR HNO3 retrievals, irrespective of retrieval approach, can never match the accuracy

and precision of those retrieved by the limb-viewing MLS. As we will demonstrate later, MLS observes a much stronger HNO3 spatial feature inside the Arctic vortex throughout the season because of its ability to measure stratospheric HNO3 minima/maxima in much narrower pressure layers with greater sensitivity to small-scale changes. Nadir IR sounders, on the other hand, are sensitive to lower stratospheric HNO3 across a single broad pressure layer, which limits its ability to observe

localized minima/maxima. But what nadir IR sounders lack in stratospheric vertical resolution, they more than make up in their ability to broadly measure significant changes in stratospheric $HNO_3$ across ~2000 km wide swaths with at least two orbital repeat cycles in the low latitudes and as much as 14 in the high latitudes.

The motivation for the work presented here is based on the fact that the Aura spacecraft carrying MLS is currently scheduled to be decommissioned within the next year. At that stage, the scientific community will lose a critical source of observations of stratospheric $O_3$ and the chemical and physical processes affecting it. While a nadir IR $HNO_3$ product cannot continue the MLS record, it can help fill the inevitable data gap until a next-generation space-based MLS-like observing capability is restored. We envisage that CLIMCAPS soundings could prove useful in monitoring polar processes with an $HNO_3$ product indicating polar stratospheric cloud (PSC) formation. Additionally, the day-to-day variations (relative changes) in CLIMCAPS $HNO_3$ abundances over the course of a season, as well as anomalies from a long-term climatology, might convey meaningful information even if the absolute magnitudes are biased. In this sense, observations from nadir-IR sounders may have a place alongside those from OMPS/LP (Ozone Mapping and Profiler Suite Limb Profiler; Flynn et al., 2006) to characterize future seasonal $O_3$ processes in the extratropical stratosphere (Wargan et al., 2020). OMPS/LP was first launched on SNPP and will continue on JPSS+ alongside CrIS. OMPS/LP is similar to MLS in that it makes high-vertical-resolution limb measurements, but instead at much shorter wavelengths in the ultraviolet (UV) to near-IR range. Unlike CrIS, OMPS/LP depends on reflected sunlight for all observations and lacks any sensitivity to stratospheric $HNO_3$. OMPS/LP primarily observes daytime PSCs, other aerosols, and $O_3$ in the upper troposphere/lower stratosphere (UTLS). Despite the benefits offered by its limb-viewing geometry, OMPS/LP has no ability to observe atmospheric conditions during the dark polar winters when stratospheric vortices typically form. This means that, without MLS, the future of stratospheric $O_3$ studies will depend on observations from an ensemble of sources, which may routinely include both CLIMCAPS and OMPS/LP soundings.

The CLIMCAPS V2.1 product is publicly available at NASA GES DISC (Goddard Earth Science Data and Information Service Center) for Aqua (2002–2016; Smith, 2019a), SNPP (2016–2018; Sounder SIPS and Barnet, 2020c) and JPSS-1 (2018–present; Smith, 2019b). Our work in this paper will determine the system upgrades for a future CLIMCAPS V3 release. We present CLIMCAPS V3 improvements for $T_{air}$ and $O_3$ in a different paper (Smith and Barnet, 2025) and focus our discussion here on the retrieval of stratospheric $HNO_3$.

In Section 2, we present our scientific rationale. Section 3 outlines the CLIMCAPS retrieval approach, which we contrast with the Bayesian Optimal Estimation (OE) framework put forward by Rodgers (2000) that has been widely adopted in many retrieval systems, including the one used for MLS and FORLI. In practice, however, the implementation of Rodgers (2000) OE varies greatly as considerations are made for different instruments and target retrieval parameters. Needless to say, a detailed comparison of retrieval systems is beyond the scope of this paper, but by making this distinction between CLIMCAPS and the generalized Rodgers (2000) framework, we aim to clarify how the CLIMCAPS system design differs from this theoretical standard and thus allows the separation of tropospheric $HNO_3$ from stratospheric $HNO_3$ during measurement inversion. We characterize five CLIMCAPS configurations for retrieving stratospheric $HNO_3$ in Section 4 and conclude by

identifying a preferred configuration for future implementation, which we showcase as a series of maps through the northern winter/spring of 2019/2020, when the Arctic vortex was particularly large and strong. We display the MLS HNO$_3$ product (Version 5) against the experimental CLIMCAPS HNO$_3$ retrieval to highlight some of the fundamental differences between these two observing systems and make the case for how a nadir-IR product such as CLIMCAPS may contribute to stratospheric polar studies in future. We summarize our recommendations for future upgrades to the CLIMCAPS HNO$_3$ product in Section 5.

## 2 Scientific Rationale

The characterization and monitoring of chemical O$_3$ loss is one of the primary applications for satellite retrievals of stratospheric HNO$_3$. In short, HNO$_3$-containing and ice PSCs initiate the chemical reactions that lead to O$_3$ loss inside polar vortices (e.g., Solomon, 1999). In addition to playing a key role in the conversion of stratospheric chlorine from benign into reactive O$_3$-destroying forms, HNO$_3$ is also involved in the deactivation of chlorine into reservoir forms at the end of winter. Therefore, the degree of chemical O$_3$ loss within any given polar vortex depends on the temperature and the presence of HNO$_3$. If PSC particles grow large enough for efficient sedimentation, then HNO$_3$ can be removed irreversibly from parts of the lower stratosphere (known as denitrification). It is not just the absolute temperature but also the thermal history of an air parcel that affects the state of PSCs and thus the rate of denitrification (Lambert et al., 2016; Murphy and Gary, 1995; Toon et al., 1986). These are complex chemical processes that can be monitored only with an ensemble of observations covering the chemical (e.g., HNO$_3$ and O$_3$) and physical (e.g., T$_{air}$ and PSCs) state of the atmosphere.

Nadir-IR sounders measure HNO$_3$ with peak sensitivity in the lower stratosphere (30–90 hPa), which is also where PSCs mainly form (e.g., Tritscher et al., 2021). Moreover, AIRS, CrIS and IASI (all in low-Earth orbit) make their measurements in wide swaths (~2000 km) with polar crossings every ~90 min irrespective of sunlight. In other words, not only do nadir-IR sounders observe HNO$_3$ in the area of interest for O$_3$ monitoring, but they do so with the ability to generate full-cover maps, both daytime and nighttime. Ronsmans et al. (2018) illustrate this with global maps of FORLI HNO$_3$ retrievals.

In Figure 1, we highlight some of the CrIS spectral features to emphasize its sounding capability. CrIS measures the IR spectrum in three distinct bands: longwave IR (650–1095 cm$^{-1}$), midwave IR (1210–1750 cm$^{-1}$), and shortwave IR (2155–2550 cm$^{-1}$). We limit our focus in Figure 1 to the two CrIS bands that report significant sensitivity to HNO$_3$, namely the longwave and midwave bands. The absorption features in Figure 1 were calculated as the absolute difference in brightness temperature (delta-BT) [Kelvin] given a perturbation of the target variable. We used the Stand-alone AIRS Radiative Transfer Algorithm (SARTA; Strow et al., 2003a) to calculate these delta-BT spectra with a CLIMCAPS sounding (i.e., the full suite of retrieved variables) as the background atmospheric state. Each target variable was perturbed in the lower stratospheric pressure layers (30–90 hPa), while keeping its value in all other layers constant.

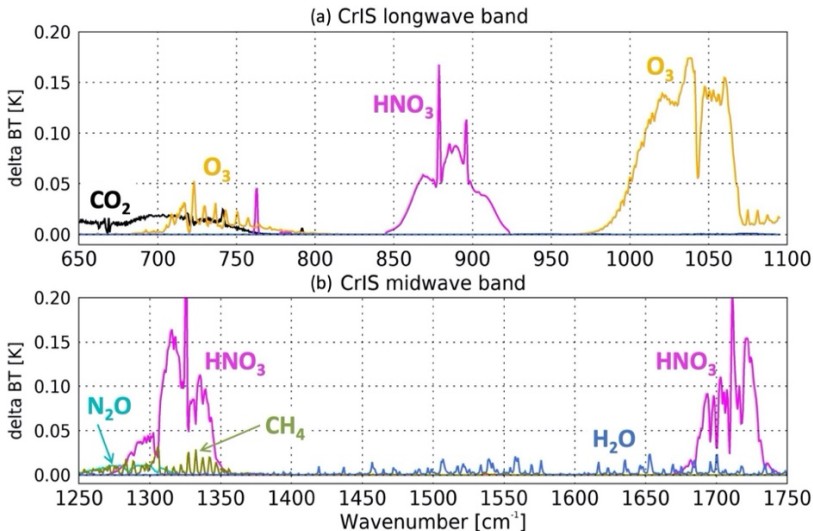

**Figure 1: Lower stratospheric (30–90 hPa) trace gas absorption spectra as the absolute delta brightness temperature (BT) for six atmospheric gases active in the thermal IR as measured by CrIS in the (a) longwave (650–1095 cm⁻¹) and (b) midwave (1210–1750 cm⁻¹) bands. These delta-BT spectra were calculated using the SARTA forward model and CLIMCAPS L2 retrievals as state parameters on 2019/12/03 at [76.7°N, 124.1°E]. Each gas was perturbed by a fraction in the stated pressure layers as follows: $H_2O_{vap}$ by 10%, $O_3$ by 10%, $HNO_3$ by 40%, $CH_4$ by 15%, $N_2O$ by 5% and $CO_2$ by 1%. The CrIS shortwave IR band (2155–2550 cm⁻¹) is absent in this figure because neither $O_3$ nor $HNO_3$ are spectrally active in this spectral region.**

Note the three distinct absorption signals for $HNO_3$ (centred about 890 cm⁻¹, 1325 cm⁻¹ and 1725 cm⁻¹) and two for $O_3$ (centred about 725 cm⁻¹ and 1025 cm⁻¹). Signals for $HNO_3$ and $O_3$ in the longwave band (Figure 1a) are in the spectral window regions ~850–900 cm⁻¹ and 1000–1050 cm⁻¹, respectively, which means that these channels have strong sensitivity to the thermodynamic structure of the troposphere, i.e., clouds, surface emissivity and $T_{air}$. The $HNO_3$ signal centred at 1325 cm⁻¹ in the midwave band is, in turn, weakly sensitive to lower stratospheric $N_2O$ and $CH_4$. We, therefore, regard $N_2O$ and $CH_4$ as spectral interference (or geophysical noise) in this $HNO_3$ band. Similarly, the $HNO_3$ signal centred at 1725 cm⁻¹ has weak sensitivity to stratospheric $H_2O_{vap}$. Even though observations of $H_2O_{vap}$ can be very useful in characterizing chemical processing, the midwave IR band does not have sufficient sensitivity to stratospheric $H_2O_{vap}$, relative to all sources of signal and noise, to yield stable retrievals in this part of the atmosphere. Nadir IR sounders are primarily sensitive to tropospheric $H_2O_{vap}$, which CLIMCAPS retrieves with high accuracy (Smith and Barnet, 2020, 2023a). As far as $O_3$ goes, the stratospheric signal centred at 725 cm⁻¹ is not only much weaker than the one centred at 1025 cm⁻¹, but it is also affected by $CO_2$ and $T_{air}$ errors. Even though CrIS is a nadir-viewing IR sounder with much lower vertical resolution than MLS, our aim with Figure 1 is to demonstrate the potential for optimizing CLIMCAPS $HNO_3$ and $O_3$ retrievals with respect to the number and spectral range of available IR channels.

## 3 Retrieval experiments and technical overview of CLIMCAPS

Here we describe the experimental setup for optimizing and characterizing CLIMCAPS $HNO_3$ retrievals. We give a generalized overview of the CLIMCAPS retrieval approach to clarify how we are able to separate stratospheric and tropospheric $HNO_3$, while simultaneously minimizing all known sources of geophysical noise in the longwave IR window region (~11 μm). In simplified terms, CLIMCAPS employs the Bayesian inversion equation, as popularized by Rodgers (2000), to iteratively retrieve a set of atmospheric state variables from nadir IR measurements as follows:

$$\hat{x} = x_a + (\mathbf{K}^T \mathbf{S}_\epsilon^{-1} \mathbf{K} + \mathbf{S}_a^{-1})^{-1} \mathbf{K}^T \mathbf{S}_\epsilon^{-1} [y - F(x_i) + \mathbf{K}(x_i - x_a)], \qquad\qquad 1$$

where $\hat{x}$ is the retrieved value of target variable $x$ at iteration $i+1$, $x_a$ the a priori estimate of $x$, $x_i$ the retrieved value of $x$ at iteration $i$, $\mathbf{K}$ the Jacobian of the forward model $F$, $\mathbf{S}_\epsilon$ the measurement error covariance matrix that quantifies spectral uncertainty, $\mathbf{S}_a$ the a priori error covariance matrix, $y$ the instrument measured IR spectrum, and $F(x_i)$ the TOA spectrum calculated by the forward model using a set of background atmospheric state variables.

In Eq. 1, the matrix, $\mathbf{S}_a$, regularizes the degree to which $y$ alters $x_a$ in solving for $\hat{x}$ such that the smaller the values in $\mathbf{S}_a$, the more $\hat{x}$ resembles $x_a$. Stated differently, if $x_a$ approximates the true state of $x$ at the time of measurement (i.e., low uncertainty about $x_a$, thus small $\mathbf{S}_a$), then the SNR of $y$ will be suppressed (strongly regularized) and $x_a$ will remain largely unaltered in $\hat{x}$. If, on the other hand, $x_a$ is a generalized estimate (e.g., static climatology) with no bearing on the true state of $x$ at the time of measurement (i.e., high uncertainty about $x_a$, thus large $\mathbf{S}_a$), then the SNR of $y$ will be weakly regularized and $\hat{x}$ will have

a large departure from $x_a$. This, however, does not guarantee accuracy in $\hat{x}$, because the spectral channels sensitive to $x$ measure both signal *and* noise; if a measurement with small SNR (high noise) is not sufficiently regularized then $\hat{x}$ could be dominated by errors. But to correctly define $\mathbf{S}_a$ for the sake of accuracy in $\hat{x}$, one needs knowledge of the true state of $x$ at the time of measurement, which is an onerous task given that AIRS, CrIS and IASI measure a wide range of atmospheric conditions across the globe, day and night. In practice $\mathbf{S}_a$ is often calculated offline as a generalized, statistical estimate of uncertainty

about $x_a$, which limits the accuracy of $\hat{x}$.

     CLIMCAPS differs from Eq. 1 according to the approach detailed in Susskind et al. (2003) to enable a long-term product with global coverage where $\hat{x}$ deviates from $x_a$ only when the measurement SNR is quantifiably high to ensure accuracy under a broad range of Earth system conditions. Most importantly, CLIMCAPS does not employ $\mathbf{S}_a$ as regularization term but instead regularizes Eq. 1 by diagonalizing the $\mathbf{K}^T \mathbf{S}_\epsilon^{-1} \mathbf{K}$ matrix (or, decomposition onto a set of orthogonal eigenvectors that it then

subsets, damps or filters based on the magnitude of the corresponding eigenvalues ($\lambda_j$, where $1 < j \le$ number of retrieval layers) and how they relate to a pre-determined static threshold, $\lambda_c$ (this is discussed in detail in Smith and Barnet, 2020). We derive $\lambda_c$ empirically for each target variable such that eigenvectors dominated by signal ($\lambda_j > \lambda_c$) are used in the retrieval of $\hat{x}$, while those dominated by noise ($\lambda_j < \lambda_c$) are filtered out. We should note that CLIMCAPS uses SARTA to calculate finite differencing Jacobians as the $\mathbf{K}_{k,l}$ matrix (where $1 \le k \le$ total number of channels and, $1 \le l \le 100$ SARTA pressure layers) that

it transforms onto the coarser retrieval pressure grid (on $j$ layers) in solving for $\hat{x}$. Once retrieved, CLIMCAPS transforms $\hat{x}$ back onto the 100 SARTA layers for ease of ingest into downstream applications. In each transformation, CLIMCAPS uses $\mathbf{S}_a$ in calculating the null space uncertainty as described in Smith and Barnet (2019). The course pressure layers CLIMCAPS uses for each retrieval variable were empirically derived offline as a series of overlapping basis functions to account for the strength and pressure-dependence of the measurement information content (Maddy and Barnet, 2008).

CLIMCAPS defines $\mathbf{S}_\epsilon$ as an off-diagonal matrix representing uncertainty from all sources of noise affecting $(y - F(x_i))$, not just $y$. These include, instrument noise, errors in the measurement and forward model, as well as uncertainty about the atmospheric state needed in the calculation of $F(x_i)$ at each retrieval step, which we refer to as the background atmospheric state, or $x_b$ (Smith and Barnet, 2019). In our case here, where $\hat{x}$ is HNO$_3$, $x_b$ includes T$_{air}$, H$_2$O$_{vap}$, O$_3$, CO, CH$_4$, N$_2$O, CO$_2$ as well as Earth surface temperature and emissivity. The instrument noise is random and quantified as the noise equivalent delta temperature (NEdT). The measurement error term includes the systematic, correlated errors introduced by apodization of the radiance measurement (in the case of CrIS, Smith and Barnet, 2025) and as well as the random amplification of NEdT and introduction of systematic errors due to cloud clearing (Smith and Barnet, 2023b). The errors in the forward model, $F$, is calculated empirically offline to account for all random and systematic sources affecting the accuracy of $F(x)$.

Table 1 details the eight pressure layers CLIMCAPS uses in its HNO$_3$ retrieval as the hinge points and effective averages. The CLIMCAPS HNO$_3$ a priori estimate, $x_a$, is a single, static climatological profile that it uses for all retrievals globally. It is also the same profile SARTA employs for its TOA radiance calculations, which is the one developed by the Air Force Geophysical Laboratory (AFGL; Anderson et al., 1986). The AFGL HNO$_3$ profile represents a global average ranging between 0.01–1.0 ppb in the UTLS, which is orders of magnitude smaller than the stratospheric values retrieved for HNO$_3$ in the extratropics during wintertime. Future work could focus on re-evaluating this AFGL profile for use as HNO$_3$ $x_a$, but the solution is not a simple replacement with a different estimate. As depicted in Eq. 1, $\hat{x}$ depends on adding measurement SNR to $x_a$, which means that whenever $x_a$ is high relative to the true state of $x$, $\hat{x}$ will be biased high. Given the large dynamic range of HNO$_3$ during the polar wintertime months, we argue that it is preferable for $x_a$ to be very small so that the retrieved product depict elevated HNO$_3$ values only where the true state ($x_{true}$) is high. Stated differently: when $x_{true}$ is very small then the corresponding IR measurement SNR for HNO$_3$ is very low and $\hat{x}$ will approximate $x_a$. Conversely, when $x_{true}$ is large, the measurement SNR is large and $\hat{x}$ will have a large departure from $x_a$. . This is all the more important for a target variable like stratospheric HNO$_3$ that is very difficult to represent with an accurate $x_a$ at each space and time retrieval footprint because of the lack of real-time observations. We can have confidence in CLIMCAPS HNO$_3$ retrievals because $\hat{x}$ will be low whenever $x_{true}$ is low, and $\hat{x}$ will significantly depart from $x_a$ only when the measurement SNR is large.

For the sake of demonstrating the feasibility of a nadir-IR stratospheric HNO$_3$ product in this paper, we focus our experiments on the spectral channel sets and regularization mechanism. Table 1 (column 3) lists the two channel subsets we test, with each subset centred on the 11 μm (~900 cm$^{-1}$) HNO$_3$ absorption band (Figure 1). In addition, the empirical threshold value, $\lambda_c$, that

CLIMCAPS employs for $HNO_3$ in the regularization of its inverse solution (Eq. 1) is given in Table 1, column 2. CLIMCAPS derives $\lambda_c$ at run-time using the input scalar variable, `Bmax`, as follows:

$$\lambda_c = \left[ 1 \Big/ (\text{Bmax})^2 \right]$$

**Table 1: Summary of the two CLIMCAPS $HNO_3$ algorithm components – `Bmax` and channel selection – tested in this study. Four different `Bmax` values and two different channel selections define five experimental setups in total, R1 – R5. The righthand column specifies the nine pressure hinge points and eight effective mean values of the pressure layers on which CLIMCAPS retrieves $HNO_3$.**

| | `Bmax` ($\lambda_c$) | Wavenumbers of the channel subsets used in the $HNO_3$ retrievals [cm$^{-1}$]. All channels are centred in the IR window region (~11µm) | Retrieval pressure layers, $j$ |
|---|---|---|---|
| **R1** | 1.5 (0.44) | 846.250, 847.500, 851.250, 855.625, 857.500, 858.125, 860.625, 861.875, 862.500, 867.500, 869.375, 873.125, 875.000, 876.875, 880.000, 881.875, 885.625, 893.125, 894.375, 895.625, 898.750, 900.000, 901.250, 902.500, 904.375, 907.500, 911.250, 912.500, 920.000 **[30 total]** | 9 x Pressure hinge points: [0.04, 9.51, 51.53, 77.24, 110.24, 170.08, 374.73, 477.96, 1042.23] hPa |
| **R2** | 0.5 (4.00) | 858.750, 859.375, 860.000, 860.625, 861.250, 861.875, 862.500, 863.125, 863.750, 864.375, 865.000, 865.625, 866.250, 866.875, 868.125, 868.750, 869.375, 870.000, 870.625, 871.250, 871.875, 872.500, 873.125, 873.750, 874.375, 875.000, 875.625, 876.250, 876.875, 877.500, 878.125, 878.750, 879.375, 880.000, 880.625, 881.250, 881.875, 882.500, 883.125, 883.750, 884.375, 885.000, 885.625, 886.250, 886.875, 887.500, 888.125, 888.750, 889.375, 890.000, 890.625, 891.250, 891.875, 892.500, 893.125, 893.750, 894.375, 895.000, 895.625, 896.250, 896.875, 897.500, 898.125, 898.750, 899.375, 900.000, 900.625, 901.250, 901.875, 902.500, 903.125, 903.750, 904.375, 905.000, 905.625, 906.250, 906.875, 907.500, 908.125, 908.750, 909.375, 910.000, 910.625, 911.250, 911.875, 912.500, 913.125, 913.750, 914.375, 915.000, 915.625, 916.250, 916.875, 917.500, 918.125, 918.750, 919.375, 920.000, 920.625, 921.250, 921.875, 922.500, 923.125, 923.750, 924.375, 925.000, 925.625, 926.250, 926.875, 927.500, 928.120 **[111 total]** | 8 x Effective mean pressure values: [1.31, 22.25, 58.52, 86.33, 129.64, 246.56, 407.27, 733.88] hPa |
| **R3** | 1.5 (0.44) | | |
| **R4** | 2.5 (0.16) | | |
| **R5** | 10 (0.01) | | |

Both CLIMCAPS and the real-time NUCAPS system have their origins in the heritage AIRS retrieval system (Susskind et al., 2003) and share many algorithm components, as described in Berndt et al. (2023). Historically, $HNO_3$ was added to the NUCAPS retrieval sequence primarily to improve the $T_{air}$ SNR (Figure 1 in Berndt et al., 2023 summarizes this step-wise approach). CLIMCAPS V2.1 takes a similar stepwise retrieval approach, as illustrated in Smith and Barnet (2023a). At every retrieval footprint, CLIMCAPS calculates an averaging kernel (AK) matrix according to Eq. 2 in Smith and Barnet (2020)

where each row, or AK, quantifies the SNR of the retrieval system for a target variable about each pressure layer. Stated differently, we can interrogate the AKs to determine the degree to which the SNR at a specific pressure layer is correlated across all other layers. For each $HNO_3$ retrieval with eight pressure layers, CLIMCAPS, therefore, generates an 8 x 8 AK matrix, or eight AKs across eight pressure layers. Figure 2 depicts the eight AKs of an $HNO_3$ retrieval on 2 February 2020 at [70.9˚N, 80.9˚E] for each of the five experimental configurations (Table 1). We should note that CLIMCAPS calculates TOA

radiance spectra with SARTA, which requires the input state variables to be defined on 100 pressure layers (Strow et al., 2003). These are also the layers on which we report the Level 2 product (Smith and Barnet, 2023a) for ease of ingest into applications downstream. At run-time, however, the profile retrievals are performed on a reduced set of broad pressure levels to more closely represent the information content the measurements have for each target variable (Maddy and Barnet, 2008).

Figure 2 shows that the CLIMCAPS HNO$_3$ AKs have relatively sharp peaks in the lower stratosphere (~50–90 hPa). By comparison, the FORLI HNO$_3$ AKs (see Figure 2 in Ronsmans et al., 2016) are smooth curves with broad peaks (~10–700 hPa) that span the mid-troposphere to stratosphere. The difference in AKs between these two nadir-IR retrieval systems tells us how each system quantifies and harnesses the SNR of the nadir IR measurements with respect to the target variable; FORLI HNO$_3$ has significant correlation across most of the retrieval layers (which is why they aggregate their product into a single value spanning the stratosphere and troposphere), while CLIMCAPS HNO$_3$ correlates predominantly across the lower stratospheric layers (which is why we can propose a stand-alone stratospheric product in this paper). Looking more closely at the five experimental configurations in Figure 2, we see that the **R1** and **R2** configurations have AK values approximating zero in the troposphere (i.e., pressures > 100 hPa) with peaks at ~70 hPa, which is favorable for retrieving lower stratospheric HNO$_3$ because it means no correlation with tropospheric SNR. However, compared to **R3**–**R5**, the **R1** and **R2** stratospheric SNR is very low overall and thus not ideal. The **R5** configuration, in contrast, presents AKs with peaks in both the mid-troposphere and the lower stratosphere, which means that their SNR about HNO$_3$ is strongly correlated across the troposphere and stratosphere. For a stand-alone stratospheric HNO$_3$ product, we are interested in a system configuration yielding AKs with distinct, large (relatively speaking) peaks centred about ~70 hPa and approximating zero in the troposphere, which is why **R3** and **R4** are attractive options for a future CLIMCAPS HNO$_3$ product. We explain this in more detail later.

In Figure 2, we also plot the AK matrix diagonal vector (AKD) as a dashed red line for **R1**–**R5**. The AKD captures the maximum of each AK and is an effective way to summarize the SNR quantified by the AK matrix. We can then aggregate any number of AKDs statistically as a mean profile with standard deviation error bars (e.g., Figure 3) for an estimate of system performance within a study region and the degree to which its SNR varies across space and time (Smith and Barnet, 2020, 2025).

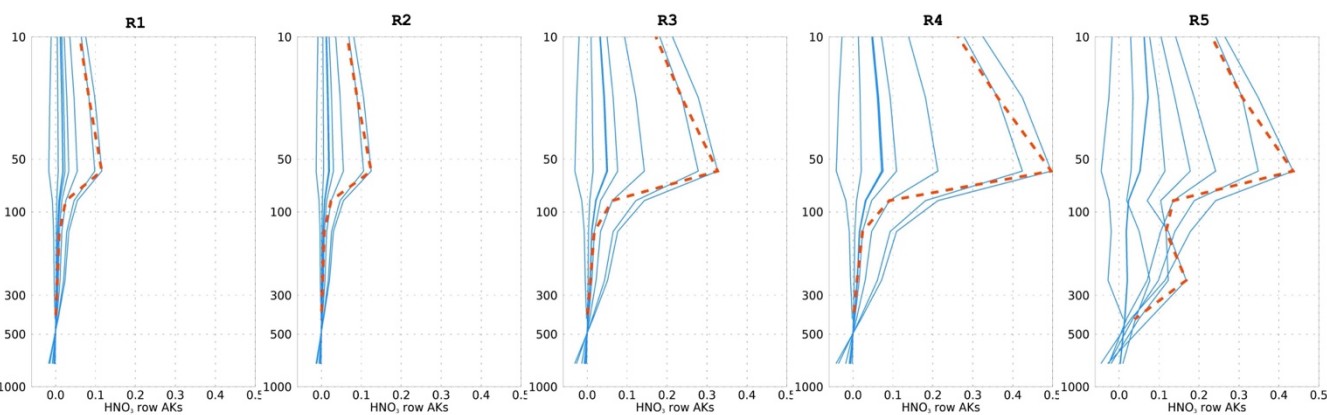

**Figure 2: The eight CLIMCAPS averaging kernel (AK) functions (blue lines) for an HNO$_3$ retrieval on 2 February 2020 at 70.9˚N, 80.9˚E according to each of the five experimental configurations, (left) R1 to (right) R5. CLIMCAPS retrieves HNO$_3$ on eight broad pressure layers as defined in Table 1. The dashed red line is the diagonal vector of the 8 x 8 AK matrix to summarize the peak SNR of each AK function.**

An AKD with values significantly greater than zero across two or more pressure layers indicate a system's ability to retrieve the target quantity across those pressure layers. The AKD error bars, on the other hand, can be interpreted as a system's sensitivity to variations in conditions at the time of measurement. However, as discussed in more detail elsewhere (Smith and Barnet, 2025), larger AK peaks and/or variation do not necessarily imply a better system or more accurate retrieval since the exact quantification of measurement SNR under all conditions is very difficult; signal may be misinterpreted as noise and vice versa. The best one can do when designing a Bayesian retrieval system is establish a SNR that yields the desired results with adequate accuracy under most conditions.

As stated earlier, Table 1 lists the two experimental $HNO_3$ channel sets used here. Experimental run **R1** represents the operational CLIMCAPS V2.1 set-up with `Bmax` = 1.5 and a 30-channel subset selected from the longwave IR window band (850–920 cm$^{-1}$, or ~11 μm). The only difference between **R1** and the operational V2.1 product is that the forward model (or rapid transmittance algorithm) error spectrum (RTAERR) is set to zero in **R1**. As identified in Smith and Barnet (2025), the V2.1 RTAERR for the CLIMCAPS-CrIS retrieval configuration is too high relative to the CLIMCAPS-AIRS configuration, and we made the recommendation that a future CLIMCAPS V3 release should update the RTAERR accordingly. An overestimated RTAERR lowers retrieval SNR by damping measurement signal, whereas an underestimated RTAERR destabilizes the SNR by allowing forward model noise to be interpreted as signal during retrieval. SNR can be destabilized when the noise is high relative to the signal, or when the noise fluctuates dramatically relative to the signal from scene-to-scene. Additionally, SNR can be destabilized when the noise (random and systematic) is not well-characterized and quantified such that it is wrongly interpreted as signal instead. Similarly, SNR can be destabilized when signal is wrongly interpreted as noise. Here, we simply set RTAERR = 0 for all five experimental runs to avoid measurement damping for the sake of illustrating the CLIMCAPS $HNO_3$ retrieval approach. An operational configuration would require the accurate representation of RTAERR across the full IR spectrum to account for errors introduced by SARTA; however, previous experience suggests that the magnitude of RTAERR is very small and on the order of the instrument noise for most channels. So, while RTAERR = 0 is technically an under estimation, it is close in magnitude to the real RTAERR and therefore not destabilizing. Historically, RTAERR was installed as an attempt to lower the weight of channels that had poor spectroscopic laboratory measurements or a large RTAERR value. In 1995, in the pre-launch AIRS era, this term was expected to be rather large — on the order of ~1° K for many channels — especially in the water band region. After the AIRS launch, the RTA fitting procedure was improved and more recent laboratory spectroscopy measurements were incorporated so that over time, the RTAERR term was reduced to very low values (<0.01° K for most channels). We now simply ignore the few remaining channels that have high RTA errors so that setting RTAERR = 0 is no longer deemed an issue for stability. The $HNO_3$ channel subset used in **R2**– **R5** employs all available CrIS channels in the ~11 μm band to maximize $HNO_3$ SNR (111 channels in total). Note that we avoid selecting channels from the two other $HNO_3$ absorption bands (Figure 1) because they are spectrally more complex, with multiple sources of interfering signals that manifest as geophysical noise in the retrieval. The IR window region, on the other hand,

provides a strong SNR for stratospheric $HNO_3$ because the predominant source of geophysical noise is from the troposphere, which CLIMCAPS separates out ahead of the inversion step.

We varied `Bmax` in the **R2**, **R4** and **R5** experimental runs from low to high: 0.5, 2.5 and 10.0, respectively (Table 1). Eigenvalues are a dimensionless quantity that represents the $HNO_3$ SNR. CLIMCAPS uses $\lambda_c$ in its regularization of the Bayesian inverse solution at run-time (Eq. 1). A measurement eigenfunction is used without *any* damping (i.e., determined 100% in the retrieval) when its eigenvalue ($\lambda_j$) exceeds $\lambda_c$. In contrast, the eigenfunctions for which $\lambda_j < \lambda_c$ are either regularized (damped) or removed from the solution space altogether (filtered). The regularization factor, `Rfac`$_i$, determines how the eigenfunctions are treated as follows:

$$\text{Rfac}_j = \frac{\lambda_j}{(\lambda_j + \Delta\lambda)}, \text{ where } \Delta\lambda = \sqrt{\lambda_c}\sqrt{\lambda_j} - \lambda_j \qquad\qquad \textbf{2}$$

When $0.05 <$ `Rfac`$_j < 1.0$, the corresponding eigenfunction is damped anywhere between 0.01% to 95% as determined by the relation $100.0(1 - $`Rfac`$_j)$. All eigenfunctions for which `Rfac`$_j \leq 0.05$ are simply removed (or damped 100%) because the assumption is that these high-frequency eigenfunctions are dominated by noise relative to the target variable. In summary, a high `Bmax` corresponds to a low $\lambda_c$, which means that `Rfac`$_j$ will be higher overall with a larger set of eigenfunctions satisfying the conditions $\lambda_j > \lambda_c$. If `Bmax` is too low for a target variable, then the measurement signal may be overdamped such that the top eigenfunctions all have $\lambda_j \ll \lambda_c$. Conversely, if `Bmax` is too high, then measurement noise may be underdamped, with higher-order eigenfunctions – that contain mostly noise relative to the target variable – contributing to the solution. In such cases, the retrieval SNR is destabilized, which can result in retrievals that do not converge or AKs that are misshapen. We consider `Bmax` to be optimized for a target variable when it enables CLIMCAPS regularization to effectively function as a noise filter during run-time. It is worth emphasizing that these eigenfunctions represent the orthogonal vectors of the measurement SNR matrix, $\mathbf{K}^T\mathbf{S}_\epsilon^{-1}\mathbf{K}$, where $\mathbf{S}_\epsilon$ represents all errors in $(y - F(x))$ as discussed earlier.

Table 2 illustrates how `Bmax` works in practice for seven profile retrievals – $T_{air}$, $H_2O_{vap}$, $CO_2$, $O_3$, $CH_4$, CO and $HNO_3$ – given the **R4** CLIMCAPS configuration. Tabulated like this, it becomes clear how the measurement signal for $T_{air}$, $H_2O_{vap}$, $CO_2$, $O_3$ and $CH_4$ is spread across multiple eigenfunctions and, in contrast, concentrated into a single dominant eigenfunction for CO and $HNO_3$. All eigenfunctions with `Rfac` values less than 5% are removed (filtered) from the solution space and thus treated as containing only noise relative to the a priori estimate. The measurement SNR matrix, $\mathbf{K}^T\mathbf{S}_\epsilon^{-1}\mathbf{K}$, projected onto an orthogonal vector space like this captures the signal by the first few eigenfunctions, while the noise separates out into the remaining eigenfunctions. Looking more closely at $H_2O_{vap}$, we see that EF1 is fully determined in the retrieval (no damping), while EF2 is damped 55%, EF3 82%, EF4 87% and EF5 93%. It is different for $T_{air}$, for which EF1 through EF8 are all damped to some degree, with EF1 9% and EF8 95% as the extremes. When we look at $O_3$, we see by far the highest EF1 eigenvalue of all the variables listed. This can be explained by the fact that nadir-IR sounders have very strong sensitivity to stratospheric $O_3$ in the 1000–1100 cm$^{-1}$ band (Figure 1) and almost negligible sensitivity to other background state variables in this spectral region.

In addition to a high SNR for EF1, the $O_3$ EF2 and EF3 have eigenvalues that are relatively high as far as nadir-IR trace gas signals go, indicating the presence of a tropospheric $O_3$ signal. As demonstrated in Smith and Barnet (2020, 2025), the nadir-IR sounder sensitivity to tropospheric $O_3$ is high enough to warrant a profile product, which Gaudel et al. (2024) evaluated as part of the Tropospheric Ozone Assessment Report 2 (TOAR-II).

There is a direct relation between CLIMCAPS regularization and the retrieval AKs. A system that is over-regularized (i.e., where signal is quantified as noise) will have AK peaks approaching zero, and a system that is under-regularized (i.e., where noise is quantified as signal) will have AK peaks approaching one. All measurements have some degree of noise, so AK peaks typically range between zero and one (Smith and Barnet, 2020). CLIMCAPS has a comprehensive error (and uncertainty) quantification scheme that accounts for the manner in which prevailing atmospheric and Earth surface conditions affect measurement SNR. This means that CLIMCAPS AKs can be used as a diagnostic metric in the analysis of system performance and the interpretation of product differences in inter-comparison studies. Another useful metric is the degrees of freedom for signal (DOFS) that is calculated as the trace of the AK matrix (or sum total of the AKD) to indicate the number of independent pieces of information that can be retrieved about the target variable, given the measurement SNR at a specific space and time location. DOFS = 1 means a single distinct quantity, DOFS = 2 means two distinct quantities, and so on. However, nadir-IR retrieval DOFS are rarely whole numbers, and the signal available for a target variable about a pressure layer is never noise-free, so in practice DOFS is a fractional value. Table 2 summarizes the CLIMCAPS DOFS for seven retrieval variables given the **R4** configuration on 2 February 2020. Note how the DOFS for $T_{air}$, $O_3$ and $H_2O_{vap}$ all exceed one, indicating measurement SNR for multiple atmospheric layers. In contrast, the DOFS for $CO_2$, $CH_4$, CO and $HNO_3$ are all below one. Upon closer inspection, we can see that the SNR (given by the eigenvalues, $\lambda_j$) for $CO_2$ and $CH_4$ are spread across multiple eigenfunctions, while CO and $HNO_3$ depend on the SNR from a single eigenfunction. This indicates that the information for $CO_2$ and $CH_4$ is not only low but also spread across multiple pressure layers, which is why we recommend integrating the retrievals into total column values ahead of their use in applications (Frost et al., 2018; McKeen et al., 2016; Smith et al., 2021). The SNR for CO and $HNO_3$, in contrast, is concentrated into a single eigenfunction, which indicates that the retrieved quantity is concentrated in a single pressure layer; as seen in the $HNO_3$ AKs (Figure 2), it is centred in the lower stratosphere.

**Table 2: Eigenvalues and DOFS from an R4 run for seven retrieval variables – $T_{air}$, $H_2O_{vap}$, $CO_2$, CO, $CH_4$, $O_3$ and $HNO_3$ – at a single retrieval footprint. Row 1 reports the static `Bmax` threshold (and corresponding $\lambda_c$ value) that CLIMCAPS employs at run-time to determine the degree of regularization for the Bayesian inverse solution. Row 2 details the eigenvalues ($\lambda_j$) and corresponding regularization factor, `Rfac_j`, of the top eight eigenfunctions (EF1–EF8). No eigenvalues with `Rfac_j` < 5% are considered in the retrieval (or damped 100%). The bottom row reports the DOFS calculated as the sum total of all `Rfac_j` values.**

| | $T_{air}$ | | $H_2O_{vap}$ | | $CO_2$ | | $O_3$ | | $CH_4$ | | CO | | $HNO_3$ | |
|---|---|---|---|---|---|---|---|---|---|---|---|---|---|---|
| `Bmax` | 0.175 | | 0.4 | | 0.35 | | 1.0 | | 1.25 | | 1.85 | | 2.5 | |
| $\lambda_c$ | 32.65 | | 6.25 | | 6,93 | | 1.0 | | 0.64 | | 0.3 | | 0.16 | |
| | $\lambda_j$ | `Rfac` | $\lambda_j$ | `Rfac` | $\lambda_j$ | `Rfac` | $\lambda_j$ | `Rfac` | $\lambda_j$ | `Rfac` | $\lambda_j$ | `Rfac` | $\lambda_j$ | `Rfac` |
| EF1 | 26.9 | 0.91 | 6.5 | 1.0 | 1.3 | 0.43 | 38.9 | 1.0 | 0.26 | 0.64 | 0.1 | 0.60 | 0.11 | 0.84 |
| EF2 | 7.8 | 0.50 | 1.2 | 0.45 | 0.45 | 0.25 | 0.8 | 0.91 | 0.006 | 0.10 | ≪0.001 | n/a | ≪0.001 | n/a |

| | | | | | | | | | | | | | | |
|---|---|---|---|---|---|---|---|---|---|---|---|---|---|---|
| EF3 | 4.6 | 0.38 | 0.2 | 0.18 | 0.16 | 0.15 | 0.14 | 0.37 | 0.002 | 0.05 | ≪0.001 | n/a | ≪0.001 | n/a |
| EF4 | 2.2 | 0.26 | 0.1 | 0.13 | 0.05 | 0.08 | 0.007 | 0.08 | ≪0.001 | n/a | ≪0.001 | n/a | ≪0.001 | n/a |
| EF5 | 1.1 | 0.19 | 0.03 | 0.07 | ≪0.001 | n/a | ≪0.001 | n/a | ≪0.001 | n/a | ≪0.001 | n/a | ≪0.001 | n/a |
| EF6 | 0.5 | 0.13 | ≪ 0.001 | n/a | ≪0.001 | n/a | ≪0.001 | n/a | ≪0.001 | n/a | ≪0.001 | n/a | ≪0.001 | n/a |
| EF7 | 0.3 | 0.09 | ≪0.001 | n/a | ≪0.001 | n/a | ≪0.001 | n/a | ≪0.001 | n/a | ≪0.001 | n/a | ≪0.001 | n/a |
| EF8 | 0.09 | 0.05 | ≪0.001 | n/a | ≪0.001 | n/a | ≪0.001 | n/a | ≪0.001 | n/a | ≪0.001 | n/a | ≪0.001 | n/a |
| DOFS | 2.49 | | 1.83 | | 0.92 | | 2.37 | | 0.79 | | 0.6 | | 0.84 | |

Another mechanism CLIMCAPS employs to maximize the SNR for each target variable is channel selection. We calculate the statistical probability of each IR channel to observe the target variable offline using the method described in (Gambacorta and Barnet, 2013). This means that the channel subsets are static across all retrieval footprints for a given product version (see Smith and Barnet, 2025 Supplement for a list of all the CLIMCAPS V2.1 channel sets). However, future system upgrades and reprocessing of the multi-decadal record can readily employ different channel subsets, so there is no requirement for the channel subsets to be fixed for all future versions of CLIMCAPS. In contrast, the degree to which the signal is captured (and noise is filtered) from the orthogonal measurement subset (eigenfunctions) is determined dynamically for each retrieval footprint at run-time. Given the changing climate, we consider this an important capability, especially for a multi-decadal product like CLIMCAPS, which cannot risk biasing its observational time-series with static assumptions about a priori uncertainty and its covariance, $\delta x_{HNO_3} \delta x_{HNO_3}^{T}$, as the regularization term. The goal of the CLIMCAPS $HNO_3$ product we propose here is to observe polar climate processes, not reflect static assumptions.

CLIMCAPS retrieves cloud top pressure and cloud fraction, but only for the troposphere, so it does not have the ability to report on PSCs that initiate heterogeneous chemical processing in the polar stratosphere. Another factor to keep in mind is that CLIMCAPS performs "cloud clearing" on each cluster of 3 x 3 instrument fields-of-view (FOVs; ~15 km at nadir) and retrieves all geophysical variables on the aggregated footprint (~50 km at nadir). Cloud clearing uses spatial information to remove the spectral cloud signal from each measurement before inversion (Smith and Barnet 2023b), and it allows CLIMCAPS to quantify (and propagate) measurement error due to clouds in all atmospheric state retrievals. This not only helps stabilize the SNR, but also affords the ability to derive meaningful quality control (QC) metrics. One of the criteria in the CLIMCAPS QC flag is to reject retrievals with high error due to cloud contamination. All CLIMCAPS retrievals, therefore, represent the atmosphere around cloud fields, not inside them.

We ran CLIMCAPS on the JPSS-1 Level 1B files of CrIS and ATMS (Advanced Technology Microwave Sounder). We refer to the experimental configuration employed in this paper simply as CLIMCAPS to indicate that the results apply to all sounder configurations in general and to distinguish it from the operational implementation of CLIMCAPS V2.1 at the GES DISC. The Level 2 retrieval files report values on the nadir-IR sounder instrument grid. There are 45 scanlines per file, and each scanline spans nearly 2000 km with 30 retrieval footprints along track, such that the spatial resolution at nadir is ~50 km and at edge-of-scan ~150 km. These satellites orbit the Earth from pole to pole, so their wide swaths have significant overlap at

high latitudes. A number of custom configurations for creating gridded Level 3 files are possible, depending on the target

application. For the sake of illustration and clarity of argument, we aggregated the CLIMCAPS retrievals from all ascending JPSS-1 orbits (~1:30 pm local equator overpass time) onto 4° equal-angle global grids. Before aggregation, we vertically integrated the $T_{air}$, $O_3$ and $HNO_3$ profiles over all pressure layers in the 30–90 hPa range. Only those retrievals that passed CLIMCAPS QC were aggregated. We added DOFS to the suite of gridded variables to help interpret the results. CLIMCAPS QC is a simple "yes/no" flag derived from a large array of diagnostic metrics that includes errors due to cloud clearing, $T_{air}$,

$H_2O_{vap}$, and cloud fraction. As our understanding of product applications matures, we envisage future CLIMCAPS upgrades to include customized QC metrics, especially for the trace gas species.

## 4 Results and Discussion

Having described the dynamic regularization mechanism of the CLIMCAPS inversion in Section 3, we now turn our attention to evaluating the five experimental runs, **R1**–**R5**. Figure 3 depicts the AKDs (mean profile and standard deviation error bars)

for all CLIMCAPS retrievals of $T_{air}$, $O_3$ and $HNO_3$ poleward of 40°N latitude on 2 February 2020. The first point to note is that the $T_{air}$ and $O_3$ AKD profiles have error bars for more pressure layers than the $HNO_3$ AKD. This is because nadir-IR sounders generally have higher information content for $T_{air}$ and $O_3$ relative to that for $HNO_3$. CLIMCAPS, accordingly, has a unique set of basis functions for each profile variable to allow the measurements to reliably map into retrieval space given the available information content. The higher the DOFS on average, the more retrieval layers are warranted. These retrieval layers

are static across all retrieval footprints.

There are five $HNO_3$ AKDs in Figure 3, corresponding to the five experimental runs. As mentioned, **R1** is closest to the CLIMCAPS V2.1 configuration, except that `RTAERR = 0`. The $HNO_3$ AKs for all five runs have peak sensitivity in the lower stratosphere, 30–90 hPa. We selected all available CrIS channels in the longwave IR window region (850–920 cm$^{-1}$) for the **R2**–**R5** CLIMCAPS runs to maximize measurement SNR for the sake of illustration. This is the same spectral region exploited

for $HNO_3$ retrievals from GLORIA (Oelhaf et al., 2019) and IASI (Ronsmans et al., 2016). With everything else held constant, the only parameter that varies among **R2** (grey), **R3** (blue), **R4** (magenta) and **R5** (gold) is `Bmax`. It is, therefore, all the more remarkable to see how the corresponding $HNO_3$ AKDs vary, not only in vertical structure, but also in their deviation about the mean in each effective pressure layer. **R2** registers the lowest values for $HNO_3$ AKs overall, and **R5** the highest. **R3** and **R4** result in AKs with similar vertical structure and variance, with **R4** having slightly higher AKD values in the middle to upper

troposphere.

In general, we know that IR sounder information content varies with ambient conditions, especially in the troposphere, where atmospheric variables have a large dynamic range in response to Earth surface and weather events (Smith and Barnet, 2019, 2020). So, we expect CLIMCAPS AKs to reflect this dynamic range with standard deviation error bars > 0.0. In general, Bayesian AKs with larger (smaller) peaks indicate an inverse solution with stronger (weaker) dependence on the measurement

relative to the a priori estimate. In principle, AK = 0 indicates that the solution is the a priori estimate, and AK = 1.0 indicates that the solution is entirely measurement-based with no dependence on an a priori estimate. But measurements contain both signal *and* noise, so neither of these extremes manifest in reality. While a strong contribution from the measurement (sharply peaked AK) could be interpreted as preferable (e.g., to compensate for a priori uncertainty), one should always keep in mind that measurements contribute noise along with signal. So, a large AK value may very well indicate that the retrieval is

dominated by measurement noise, not signal. This is why dynamic regularization of the inverse problem at run-time has proven to be such a robust mechanism for CLIMCAPS retrievals since the eigenvalue decomposition of each measurement SNR matrix helps filter noise. This simplifies error quantification during retrieval and minimizes the probability that retrievals are contaminated by measurement noise that is difficult to identify and quantify otherwise. Note that by "measurement noise", we do not simply mean the instrument error spectrum (or noise equivalent delta temperature, NEdT). Rather, measurement noise

encapsulates all spectral information not directly related to a target variable. For example, the CrIS channels sensitive to mid-tropospheric $CH_4$ are also sensitive to mid-tropospheric $H_2O_{vap}$ (Smith and Barnet, 2023a). If the target variable is $CH_4$, then $H_2O_{vap}$ should be treated as geophysical noise, and vice versa. Smith and Barnet (2019) explain how we account for the many sources of measurement noise in CLIMCAPS.

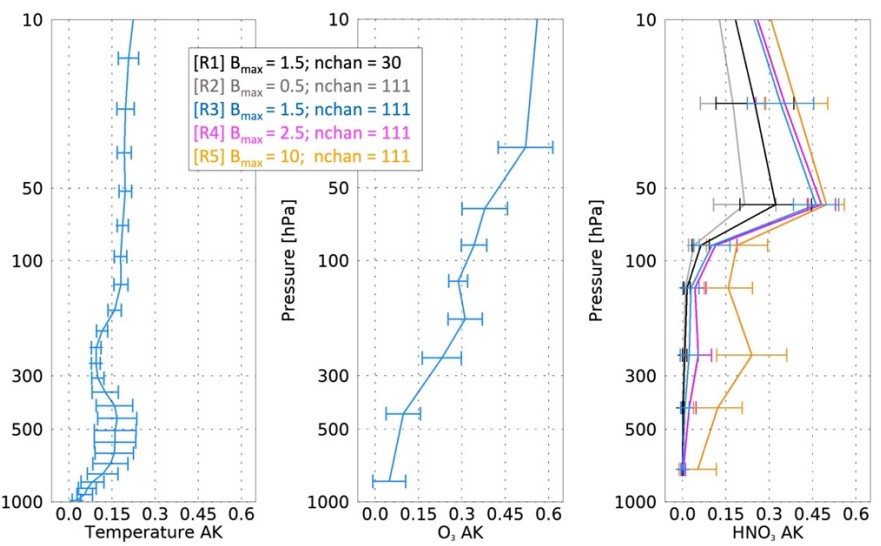

**Figure 3: A statistical summary of CLIMCAPS signal-to-noise ratio (SNR) for all the temperature (left), $O_3$ (middle) and $HNO_3$ (right) retrievals north of 40°N latitude on 2 February 2020. The profiles represent the average of the respective averaging kernel matrix diagonal vectors (AKD) with standard deviation error bars to indicate the degree to which CLIMCAPS SNR varies across retrieval pressure layers within the study region. CLIMCAPS retrieves temperature on 31 pressure levels, $O_3$ on 11 layers, and**

**$HNO_3$ on 8 layers, hence the difference in the number of error bars across the three variables.**

Table 3 tabulates the eigenvalues and regularization factor (`Rfac`$_j$) relative to $HNO_3$ for the first five eigenfunctions (EF1–EF5) of each experimental run. We report these values for two retrieval footprints independent of each other as well as the

footprint represented in Table 2. `Bmax` determines the eigenvalue threshold, $\lambda_c$. In turn, `Rfac`$_j$ depends on the corresponding

eigenvalue, $\lambda_j$, as well as $\lambda_c$. Table 3 illustrates the degree to which the HNO$_3$ eigenfunctions are regularized in the inverse solution at run-time. The EF2 eigenvalue is orders of magnitude smaller than the one for EF1, and all remaining eigenvalues (EF3–EF5) are roughly the same order of magnitude as EF2. As shown in Table 2, this is not the case for all CLIMCAPS retrieval variables. What it tells us is that most (if not all) of the signal for stratospheric HNO$_3$ compresses into a single eigenfunction that can readily be isolated from most (if not all) tropospheric signal and noise. With this table and associated

discussion, we aim to illustrate how CLIMCAPS regularization works in practice and the things we consider when planning a system upgrade. The results presented in this paper help us identify which experimental configuration to adopt, test and refine for a future V3 public release. An in-depth quantification of the eigenfunctions across all retrieval footprints, and especially the type of conditions of interest to seasonal monitoring in the extratropics, will be the focus of future work. Our objective here is simply to draw broad conclusions about each experimental configuration for the sake of identifying the path forward.

**Table 3: A summary of the top five HNO$_3$ eigenvalues for two distinct retrieval footprints as processed by each experimental configuration, `R1 – R5`. The eigenfunctions of the radiance measurements are dependent on channel selection but independent of `Bmax` ($\lambda_c$). The latter is an empirical scalar that determines the degree to which CLIMCAPS regularizes Bayesian inverse solutions.**

| | | R1 ($\lambda_c$ = 0.16) | | R2 ($\lambda_c$ = 4.0) | | R3 ($\lambda_c$ = 0.44) | | R4 ($\lambda_c$ = 0.16) | | R5 ($\lambda_c$ = 0.01) | |
| --- | --- | --- | --- | --- | --- | --- | --- | --- | --- | --- | --- |
| | | $\lambda_j$ | Rfac | $\lambda_j$ | Rfac | $\lambda_j$ | Rfac | $\lambda_j$ | Rfac | $\lambda_j$ | Rfac |
| Footprint #1 | EF1 | 0.01 | 0.15 | 0.16 | 0.20 | 0.22 | 0.70 | 0.34 | 1.0 | 0.33 | 1.0 |
| | EF2 | 0.8E-05 | n/a | 0.1E-03 | n/a | 0.2E-03 | n/a | 0.4E-03 | 0.05 | 0.6E-04 | 0.08 |
| | EF3 | 0.2E-05 | n/a | 0.2E-04 | n/a | 0.3E-04 | n/a | 0.4E-04 | n/a | 0.4E-04 | 0.06 |
| | EF4 | 0.8E-06 | n/a | 0.1E-04 | n/a | 0.1E-04 | n/a | 0.3E-04 | n/a | 0.8E-05 | n/a |
| | EF5 | 0.6E-06 | n/a | 0.3E-05 | n/a | 0.3E-05 | n/a | 0.1E-04 | n/a | 0.3E-05 | n/a |
| | | DOFS = 0.15 | | DOFS = 0.2 | | DOFS = 0.7 | | DOFS = 1.05 | | DOFS = 1.14 | |
| Footprint #2 | EF1 | 0.25 | 0.76 | 0.02 | 0.07 | 0.44 | 1.0 | 0.05 | 0.56 | 0.05 | 1.0 |
| | EF2 | 0.7E-04 | n/a | 0.2E-04 | n/a | 0.2E-03 | n/a | 0.5E-04 | n/a | 0.5E-04 | 0.07 |
| | EF3 | 0.1E-04 | n/a | 0.3E-05 | n/a | 0.4E-04 | n/a | 0.7E-05 | n/a | 0.7E-05 | n/a |
| | EF4 | 0.9E-05 | n/a | 0.2E-05 | n/a | 0.1E-04 | n/a | 0.4E-05 | n/a | 0.4E-05 | n/a |
| | EF5 | 0.5E-05 | n/a | 0.1E-05 | n/a | 0.1E-05 | n/a | 0.1E-05 | n/a | 0.1E-05 | n/a |
| | | DOFS = 0.76 | | DOFS = 0.07 | | DOFS = 1.0 | | DOFS = 0.56 | | DOFS = 1.07 | |

Comparison of the eigenvalues in Table 3 to the AKD profiles in Figure 3 illuminates the observed differences. At the extremes are **R2** (`Bmax` = 0.5) and **R5** (`Bmax` = 10.0). The former harvests signal exclusively from the first eigenfunction in both cases, while the latter does so from two or more. This difference manifests in the **R2** AKDs (grey) approaching zero in the troposphere, while the **R5** tropospheric AKDs (gold) not only exceed 0.0 by a significant margin, but they also display a large

dynamic range. Additionally, of all five runs, the **R2** AKDs register the lowest values in the stratosphere overall. We attribute

this to overdamping. With `Bmax` set to 0.5, **R2** has the highest threshold for determining whether an eigenfunction should be damped in the retrieval, with $\lambda_c = 4.0$. We have yet to observe a CLIMCAPS $HNO_3$ eigenvalue exceeding 4.0. This means that the first $HNO_3$ eigenfunction will always be damped in **R2**, irrespective of signal strength. Not only does $\lambda_c$ determine which eigenfunctions to damp, it also determines the degree to which they are damped, such that the larger the difference between $\lambda_c$ and $\lambda_j$, the smaller the `Rfac`$_j$ values, and the higher the degree of damping. **R2** illustrates how CLIMCAPS has

the ability to overdamp nadir-IR measurements to the detriment of the inverse solution. Given that our objective with CLIMCAPS is to harvest as much of the available measurement signal as possible, while simultaneously accounting for most of the known sources of measurement noise, the value assigned to `Bmax` is an important consideration.

There is more to diagnose from the results presented in Table 3. **R5** has the highest `Bmax` (lowest $\lambda_c$) of all the runs, yet in the stratosphere its AKDs approximate those from **R3** and **R4** (Figure 3). Moreover, irrespective of retrieval configuration, the

$HNO_3$ AKDs are always much smaller than 1.0, even at their peak around 50 hPa. We deduce that there must be an upper limit to the measurement signal for $HNO_3$, regardless of system parameters. In Table 3, we see that the eigenvalues of EF1 are always much less than 1.0, unlike those for $T_{air}$, $H_2O_{vap}$, $O_3$ and $CO_2$ (Table 2). This is because nadir-IR sensitivity to stratospheric $HNO_3$ is low even if the system is optimized.

When DOFS = 1.0, it means that the measurements contain one piece of information about the target variable. But this does

not imply that one piece of measurement signal perfectly maps into one piece of atmospheric retrieval during inversion. For CLIMCAPS products, it simply means that the equivalent of one eigenvector determined the inverse solution in all pressure layers at a specific retrieval footprint. One can expand this argument as follows: The **R1**–**R3** configurations rarely have DOFS exceeding one. As seen in Figure 3, these are also the AKDs with tropospheric values approaching zero. Only **R4** and **R5** have tropospheric AKDs visibly greater than zero, and they are also the only two configurations often yielding DOFS > 1.0.

This leads us to conclude that EF1 contains most of the available stratospheric signal for $HNO_3$, while EF2 and EF3 almost exclusively quantify tropospheric signal and noise. This clear distinction between EF1 and the other eigenfunctions is not the case for all retrieval variables. Table 2 illustrates that $T_{air}$, for example, depends on multiple eigenfunctions, all partially damped but none completely undamped. Wespes et al. (2007) reported that the DOFS of retrievals from IMG radiances range between 0.7 and 1.8 and concluded that this implied the ability of IMG measurements to provide two independent pieces of

$HNO_3$ information – tropospheric and stratospheric partial columns. While a similar range is recorded for IASI $HNO_3$ DOFS (Ronsmans et al., 2016; Wespes et al., 2022), the retrievals are nonetheless presented as total column values. When a $HNO_3$ retrieval system, like FORLI, regularizes its Bayesian inversion along the full atmospheric column, without the ability to filter measurement noise at run-time, the stratospheric and tropospheric retrievals are correlated because their spectral SNR is correlated. A total column is, therefore, the only way to report such a retrieval to obtain a stable product. We argue that the

CLIMCAPS approach to Bayesian inversion, on the other hand, benefits $HNO_3$ specifically because the stratospheric and

tropospheric SNR can be decomposed into two separate eigenfunctions. This, of course, does not mean that it is the only Bayesian approach with the ability to distinguish an independent piece of stratospheric $HNO_3$ information. Compared to FORLI, however, CLIMCAPS has this novel capability.

Another aspect worth noting is that both **R1** and **R3** have `Bmax` = 1.5 ($\lambda_c$ = 0.44), yet stratospheric **R1** AKDs (Figure 3, black) are significantly smaller than those from **R3** (blue). This is due to the fact that the **R1** eigenfunctions are derived from 30 nadir-IR channels and those for **R3** from 111 channels. It is tempting, therefore, to conclude that this supports the use of all available channels during retrieval, but that would be an over-simplification. Figure 1 (as well as Figure 2 in Smith and Barnet, 2020) illustrates how nadir-IR measurements are highly mixed signals of multiple Earth system variables. We also know, empirically, that measurement SNR with respect to a target variable varies substantially because it depends on ambient conditions. One can enhance the effectiveness of CLIMCAPS regularization during inversion by pre-selecting measurement subsets with a high probability of SNR > 1.0. This means selecting channels with high sensitivity and low geophysical noise (interference from background state variables) with respect to the target variable.

Figure 4 summarizes our discussion of DOFS for **R1** through **R5** with maps of the NH centred on the North Pole on 2 February 2020. The gridded average of DOFS, which we refer to as `avg(DOFS)` from here on, is displayed in the top row, with the standard deviation of the gridded DOFS, or `stdev(DOFS)`, on the bottom. **R1** and **R3** yield similar spatial patterns for `avg(DOFS)` but different patterns for `stdev(DOFS)`. With the **R1** eigenfunctions derived from 30 channels, unlike 111 in **R3**, it is possible that the smaller channel set lowers **R1** SNR to the point that its retrieval DOFS become unstable (i.e., highly variable). Given what we have learned from the values reported in Table 3, we argue that it is preferable for $HNO_3$ DOFS to approximate 1.0 – never to exceed it – and for EF1 to be fractionally damped only for the cases where the measurement SNR is low to begin with, such as where stratospheric $HNO_3$ concentrations are low. It is, therefore, interesting to compare **R3** and **R4**. Their `stdev(DOFS)` patterns are similar in that `stdev(DOFS)` is high wherever `avg(DOFS)` is low, despite the **R3** `avg(DOFS)` being much lower overall and **R4** `avg(DOFS)` approximating 1.0 across most of the study area. We can make sense of this when we revisit Tables 2 and 3, which show that $Rfac_j$ can have a large dynamic range – and thus high `stdev(DOFS)` – for small variations in all $\lambda_j < \lambda_c$. So even though **R3** may yield a relatively stable $HNO_3$ retrieval given its low `stdev(DOFS)` across most of the study region, its `avg(DOFS)` indicate that EF1, or the eigenfunction with most of the stratospheric $HNO_3$ signal, is probably overdamped. Of all five cases, we argue that **R4** yields the closest representation of what is needed for a stratospheric $HNO_3$ product aimed at science objectives related to ozone chemistry. In contrast, **R2** and **R5** epitomize what is *not* desirable in a stratospheric $HNO_3$ product, but for different reasons.

The maps in Figure 4 indicate that **R2** `stdev(DOFS)` is low wherever **R3** `stdev(DOFS)` is high (and vice versa), even though their `avg(DOFS)` maps show not dissimilar patterns. As discussed earlier, the **R2** configuration is associated with a very high $\lambda_c$, relatively speaking. And, the higher the $\lambda_c$, the larger the difference between $\lambda_{EF1}$ and $\lambda_c$, leading, in turn, to a lower $Rfac_j$ and lower retrieval DOFS overall. In fact, the **R2** difference between $\lambda_c$ and $\lambda_{EF1}$ is so large that EF1 is always strongly

damped, i.e., Rfac$_j$ is small and DOFS low. This is why the **R2** avg(DOFS) is overall significantly lower than that of all other configurations in Figure 4.

In contrast, **R5** avg(DOFS) exceeds 1.0 across the whole study area. This is not ideal because it means that the stratospheric HNO$_3$ retrieval is correlated with tropospheric noise. Two regions of the **R5** stdev(DOFS) map stand out as having values significantly higher than the background, namely the zones centred about (i) [40˚–90˚N, 50˚–120˚E], or the Siberian landmass, and (ii) [40˚–60˚N, 40˚–160˚W], which is North America. The Earth surface, boundary layer conditions and tropospheric weather processes are highly variable over land. When HNO$_3$ DOFS > 1.0, the measurement SNR for tropospheric conditions

contributes to the retrieval as higher-order eigenfunctions that are damped anywhere between 0.01% to 95% according to Rfac$_i$. This leads us to conclude that it may be worth considering a customized HNO$_3$ configuration for future CLIMCAPS upgrades. For example, the Rfac of EF2 (and all higher-order eigenfunctions) can be set to a static value ($\leq 0.05$) to always filter higher-order eigenfunctions, except EF1, to cap DOFS at 1.0 and thus isolate the stratospheric HNO$_3$ SNR. Of course, more research would need to be done to determine if such an approach is feasible under all conditions (i.e., demonstrate EF2

is always tropospheric), but CLIMCAPS could support such a customization in principle.

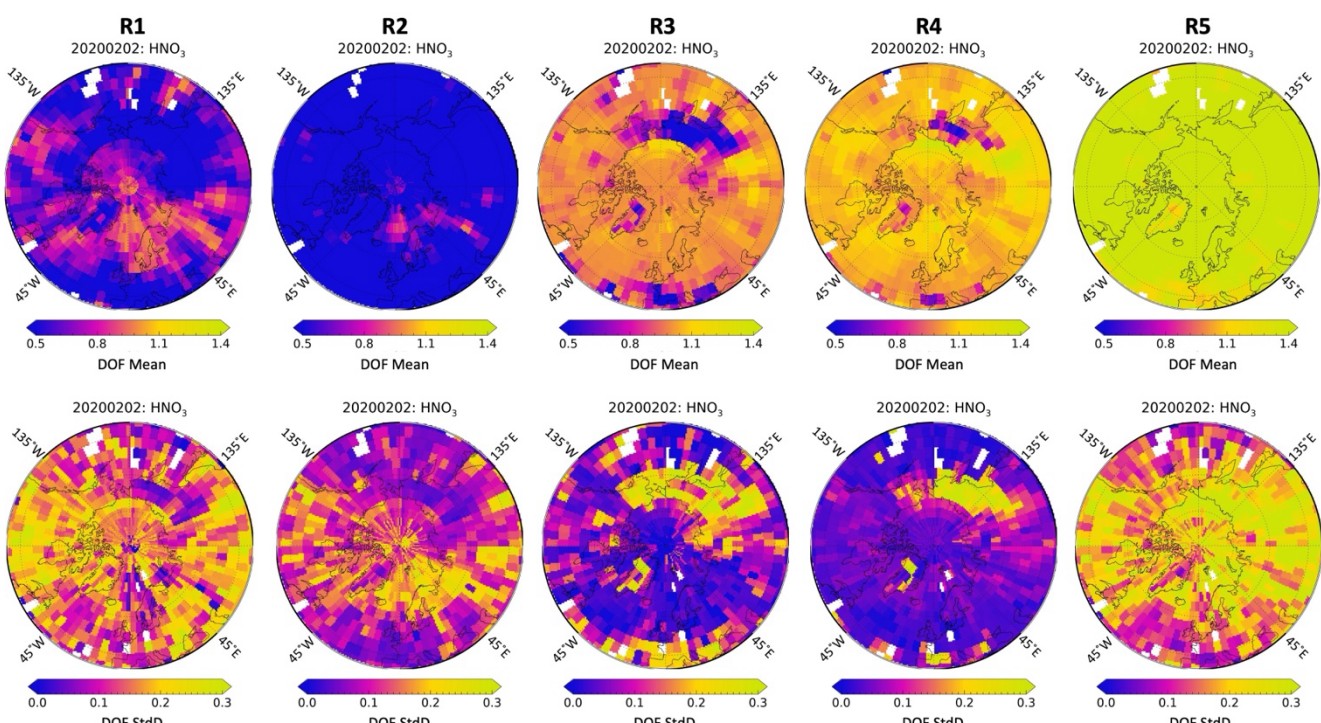

**Figure 4: CLIMCAPS information content metrics (or, DOFS) aggregated onto a 4˚ equal-angle grid poleward of 40˚N latitude. (Top row) Averaged HNO$_3$ DOFS. (Bottom row) Standard deviation of the HNO$_3$ DOFS. (Left to right) The five experimental setups as defined in Table 1.**

We contrast the CLIMCAPS HNO₃ retrievals from the **R4** experimental configuration with those from limb-viewing MLS in Figure 5 for five days spanning the Northern Hemisphere winter of 2019/2020. CLIMCAPS HNO₃ has global coverage (90˚S to 90˚N) and is presented here as spatial averages on a 4˚ equal-angle grid. The MLS product is the Level 2 V005 HNO₃ mixing ratio (Manney, 2021) that has near-global coverage (82˚S to 82˚N) and profile retrievals spaced 1.5˚ along the orbital track with roughly 15 orbits per day. We integrated both profile products across their retrieval layers spanning 30–100 hPa for ease of comparison. This figure depicts the formation and dissipation of the Arctic vortex as illustrated by the coincident CLIMCAPS retrievals of stratospheric $T_{air}$ and $O_3$. We present CLIMCAPS V3 improvements for $T_{air}$ and $O_3$ in a different paper and focus our discussion here on the comparison between MLS and CLIMCAPS retrievals of HNO₃. It is clear that MLS observes a much stronger HNO₃ spatial feature inside the Arctic vortex throughout the winter season. Note how the spatial patterns of CLIMCAPS HNO₃ strongly align with those from MLS at the onset of the vortex in November, and again as HNO₃ reaches its first distinct seasonal feature in February. By March, however, the CLIMCAPS HNO₃ feature weakens relative to that from MLS and by April is largely absent as the temperatures in the vortex start to rise. By May, the vortex has dissipated (as seen in the CLIMCAPS temperature and O₃ maps), along with the distinct seasonal HNO₃ feature in both products. It is worth noting that CLIMCAPS HNO₃ registers a strong low-HNO₃ feature (such as that visible on 2 February 2020) only when coincident with wintertime minima in both temperature and O₃, never outside of the conditions indicating the presence of the winter polar vortex (not shown). Additionally, note the strong agreement in spatial patterning between MLS HNO₃ and CLIMCAPS O₃ throughout the season, while the same cannot be said for the colocated CLIMCAPS HNO₃ at this stage. It is worth reminding the reader here that a mature, optimal CLIMCAPS HNO₃ product does not exist yet. Figure 5 presents CLIMCAPS retrievals produced by the experimental **R4** configuration, which is only a first step towards achieving a viable stratospheric nadir IR HNO₃ product in future. It will be interesting to determine the degree to which the CLIMCAPS HNO₃ retrieval can be optimized for a better correlation with MLS HNO₃ throughout the lifetime of the Arctic vortex.

One aspect that needs further investigation is how Earth surface conditions affect the HNO₃ retrieval. The 850–900 cm⁻¹ spectral region sensitive to HNO₃ (Figure 1) is colloquially known as the IR window region because it is predominantly sensitive to Earth surface conditions, and specifically to surface emissivity and skin temperature. This means that CLIMCAPS needs to accurately account for Earth surface conditions as a source of geophysical noise during each HNO₃ retrieval. That the HNO₃ retrievals are consistently elevated over some parts of Greenland (~45˚W) relative to the surrounding HNO₃ retrievals over ocean throughout most of the season (Figure 5) is evidence that the **R4** configuration is not yet optimized. This indicates that we need to investigate how icy land surfaces are represented in the retrieval.

Compared to nadir sounders, limb sounders have the ability to observe stratospheric HNO₃ minima/maxima in much narrower pressure layers with greater sensitivity to small-scale changes. As seen in Figure 3, CLIMCAPS is sensitive to stratospheric HNO₃ across a single broad pressure layer. This reflects a fundamental limitation of nadir sounders. In fact, no matter how much we optimize the HNO₃ retrieval configuration, CLIMCAPS will never retrieve lower-stratospheric HNO₃ minima/maxima with the same accuracy as MLS. But absolute accuracy is not the only metric that determines product value

in applications. Often the spatial gradients themselves provide relevant information, as demonstrated in severe weather forecasting with gridded NUCAPS soundings (Berndt et al., 2020). Moreover, the day-to-day variations (relative changes) in $HNO_3$ abundances over the course of the season, as well as anomalies from a long-term climatology, might convey meaningful information even if the absolute magnitudes are wrong. Even though nadir-IR systems may be limited in the accuracy with which they can retrieve lower-stratospheric $HNO_3$ concentrations, they do provide a complete spatial representation day and

night, through all seasons independent of sunlight. Future work could, therefore, focus on designing a custom $HNO_3$ gridded product with quality flags and a space-time aggregate configuration that clearly delineates the target features, such as the $T_{air}$ product developed for aviation forecasts (Weaver et al., 2019). The applicability of CLIMCAPS $HNO_3$ could be significantly broadened in combination with coincident CLIMCAPS retrievals of stratospheric $T_{air}$ and $O_3$.

It is worth taking a moment to reflect on the CLIMCAPS $HNO_3$ a priori estimate. At the core of any Bayesian inverse

solution is its dependence on an a priori estimate to initialize the retrieval. CLIMCAPS uses the AFGL climatology (Anderson et al., 1986) to define a static $HNO_3$ a priori estimate for retrievals at all footprints globally. While the AFGL climatology does not represent typical $HNO_3$ concentrations in the polar regions (it is orders of magnitude smaller than what MLS measures), it does benefit the CLIMCAPS $HNO_3$ product, given our target application of extratropical heterogeneous chemical processing in the lower stratosphere. It means that one can interpret the $HNO_3$ maps in Figure 5 as representing

what the nadir sounders measure, not what the a priori estimate represents. Stated differently, CLIMCAPS depicts elevated $HNO_3$ values (i.e., retrieval > a priori estimate) only where the nadir measurements have sensitivity to $HNO_3$ due to measurable concentrations in the lower stratosphere. Conversely, the CLIMCAPS $HNO_3$ retrieval approximates the a priori estimate wherever stratospheric $HNO_3$ concentrations are too low to be measurable by the nadir-IR sounders. While such an atypical a priori estimate may contribute to a slower rate of convergence during Bayesian inversion, it does not significantly

impact the CLIMCAPS retrieval, which routinely logs rapid convergence (2–3 iterations) to a stable solution because of how it employs various compression techniques, such as eigenfunction-based regularization and projection of the atmospheric profile onto a reduced set of broad pressure layers. The result is a retrieval product that depicts $HNO_3$ spatial patterns as a function of measurement information content, or nadir sounder observing capability. One could argue that a larger a priori estimate for $HNO_3$ would benefit the polar $HNO_3$ retrievals, but CLIMCAPS is a multi-user, global product suite, and we

would need to carefully consider the impact of such a change to the $HNO_3$ a priori estimate on the retrieval suite as a whole since CLIMCAPS retrieves its Earth system parameters in series (see Figure 3 in Smith and Barnet, 2023a).

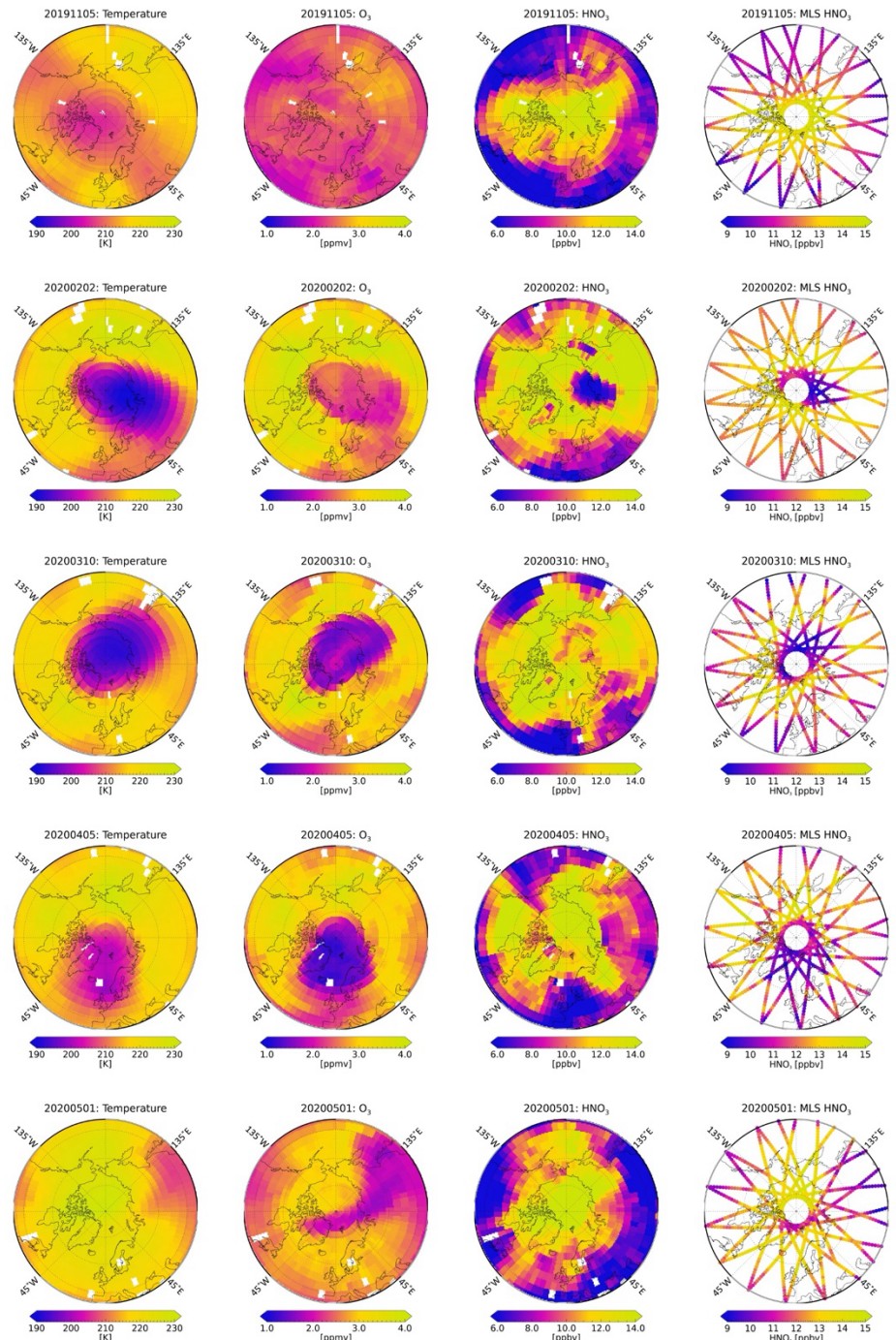

**Figure 5: A time series of lower stratospheric CLIMCAPS retrievals of $T_{air}$, $O_3$, and $HNO_3$ produced by the R4 experimental configuration aggregated onto a 4° equal-angle grid throughout the 2019/2020 Northern Hemisphere winter season poleward of 40°N latitude. The column on the right represents the MLS V5 Level 2 profile $HNO_3$ retrievals at 1.5° intervals. Both the CLIMCAPS and the MLS profile products were vertically integrated across their respective retrieval layers spanning 30–100 hPa.**

The CLIMCAPS retrievals of $T_{air}$, $H_2O_{vap}$ and $O_3$ need to serve a broader range of scientific foci with more accurate estimates of absolute quantities, so for those variables Smith and Barnet (2019, 2020) instead implemented as a priori estimate a

640 reanalysis model, specifically, the Modern-era Retrospective analysis for Research and Applications (MERRA-2; Gelaro et al., 2017). In future, and as our knowledge of product applications evolves, we can test the sensitivity of CLIMCAPS $HNO_3$ retrievals to different a priori estimates, such as zonal estimates based on MLS retrievals or the FORLI $HNO_3$ a priori estimate, which is an aggregate profile derived from chemistry transport model fields and other retrieval systems (Hurtmans et al., 2012).

## 5 Conclusions

Nadir-IR measurements, like those from AIRS, CrIS and IASI, have sensitivity to lower stratospheric (30–90 hPa) $HNO_3$ in the ~11 μm window region (850–920 cm$^{-1}$) of their longwave IR bands. This paper provided a progress report on the development of a stratospheric $HNO_3$ product for the observation of ozone loss in the extratropics. We demonstrated how stand-alone estimates of stratospheric $HNO_3$ can be retrieved from nadir IR measurements such that they are largely uncorrelated with the coincident tropospheric SNR in the measurements. This is achieved by decomposing the set of $HNO_3$

sensitive channels into orthogonal vectors that function to isolate the stratospheric $HNO_3$ SNR from most of the tropospheric SNR that would otherwise correlate with the stratospheric HNO3 retrieval.

The MLS on Aura is scheduled to be decommissioned in 2026, terminating the state-of-the-art $HNO_3$ dataset critical to the monitoring and scientific understanding of processes governing extratropical ozone. The goal of this paper is to demonstrate how stratospheric $HNO_3$ can be retrieved from nadir IR measurements such that the retrieved information is largely

independent of coincident tropospheric SNR.

The only other $HNO_3$ IR product in operation today is from FORLI, an OE retrieval system for IASI measurements. The FORLI product reports $HNO_3$ retrievals as total column quantities because their stratospheric and tropospheric information is correlated (Ronsmans et al., 2018). The work presented here challenges Ronsmans et al. (2016, 2018) by demonstrating that a correlated tropospheric+stratospheric $HNO_3$ retrieval from IR measurements is not inevitable; a retrieval method can be set up

such that a stand-alone stratospheric $HNO_3$ product is viable.

We used CLIMCAPS as the bedrock system for this demonstration because it allows the selection of individual eigenfunctions generated by the orthogonal decomposition of the measurement SNR matrix at run-time. We show here how the stratospheric $HNO_3$ signal measured by nadir IR sounders projects into a single eigenfunction that can be isolated from most of the tropospheric SNR otherwise coincident in the $HNO_3$-sensitive IR spectral channels.

We tested five CLIMCAPS configurations for $HNO_3$ retrievals and demonstrated how, unlike those of FORLI, the $HNO_3$ AKs can peak across lower stratospheric pressure layers and approach zero across all tropospheric layers. For this reason, the work presented here is novel and promises to improve upon the status quo by allowing a stand-alone stratospheric $HNO_3$ product from nadir IR measurements.

In a series of CLIMCAPS retrievals throughout the northern hemisphere winter of 2019/2020 using the **R4** configuration, we illustrated how CLIMCAPS $HNO_3$ compares against MLS $HNO_3$ and demonstrated that it reflects real stratospheric patterns under some conditions. Future work will investigate how CLIMCAPS $HNO_3$ can be optimized in terms of its a priori estimate and the quantification of uncertainty in background parameters, such as Earth surface temperature and emissivity, to improve the accuracy of its observation of stratospheric $HNO_3$ under a broader range of conditions.

The goal of this paper is to report on the degree to which the CLIMCAPS $HNO_3$ retrieval configuration can be improved for the purpose of reporting a stratospheric partial-column product that may prove useful in filling the data gap when Aura is decommissioned next year. Nadir-IR products alone cannot match or replace the limb-viewing MLS observing capability, but, paired with those from OMPS/LP, could help monitor extratropical processes with a $HNO_3$ product indicating polar stratospheric cloud (PSC) formation irrespective of sunlight. Overall, the work reported here clarified the steps we need to take to upgrade the CLIMCAPS $HNO_3$ product in a future release.

## Acknowledgements

We would like to acknowledge that NOAA, and specifically Walter Wolf, supported the development of the $HNO_3$ retrieval code during NUCAPS algorithm development, circa 2003 to 2008. Without this support it would have been extremely difficult to retrofit the infrastructure necessary to add the product at a later date, and it would have been prohibitive to perform the work discussed in this paper under the NASA ROSES maintenance funding alone (ROSES grant no. 80NSSC21K1959). We also acknowledge Fengying Sun for a number of early studies to optimize the $HNO_3$ retrieval step for use within the operational NUCAPS configuration. During the development of CLIMCAPS (2018–2020), we were able to leverage more than a decade of experience within NOAA. Work at the Jet Propulsion Laboratory, California Institute of Technology, was carried out under a contract with NASA (no. 80NM0018D0004). The preparation and revision of this manuscript was done in personal time due to a lack of follow-on funding for this legacy NASA and NOAA system.

## Competing interests

The contact author has declared that none of the authors has any competing interests.

## Author contribution

NS conceptualized this work and designed the experiments. CB is the architect and developer of the CLIMCAPS software. NS performed the formal analysis and generated all graphics. MLS and CB reviewed the study and provided critical input that guided the analysis and interpretation of results. NS wrote the original draft, with MLS and CB both reviewing the manuscript before submission.

## Data Availability

The CLIMCAPS V2.1 L2 retrieval product is archived at and distributed by the NASA GES DISC for Aqua (2002–2016; Smith, 2019a), SNPP (2016–2018; Smith, 2019c) and JPSS-1 (2018–present; Smith, 2019b). We additionally employed the MLS/Aura $HNO_3$ V005 Level 2 product also archived at GES DISC (Manney, 2021).

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
