# Peer review of "Mapping stratospheric nitric acid ( $\text{HNO}_3$ ) patterns in the extratropics with nadir-viewing infrared sounders – a retrieval perspective"

_EGUsphere, 2025_

## Author Response (AR1)

**Citation**: https://doi.org/10.5194/egusphere-2025-1569-RC1

[Reviewer]: The authors present a progress report on the development of the CLIMCAPS HNO3 retrieval, building on their own preceding work. Background to these efforts is the foreseeable "limb gap" caused by the imminent discontinuation of current limb sounders operation. The aim of the development is to obtain a data product that represents conditions in the stratosphere, specifically temporal and spatial patterns and developments. This goal is attempted to be achieved through the appropriate selection of spectral ranges and dynamic regularization adapted to local atmospheric conditions.

The technical part of the paper, i.e., the investigation into the selection of a suitable parameter set, is carefully presented. However, there are some weaknesses in terms of clarity and structure in the general presentation and also in the presentation and critical discussion of the comparisons with MLS data. Some aspects deserve better or more detailed presentation and discussion.

**My general comments are as follows:**

[Reviewer]: The introduction should be better structured, possibly with the help of subsections, since some of the information it contains seems scattered and disjointed. Some later, and more generic, passages could be inserted here (example see below).

[Response]: Noted. We have re-evaluated and edited the introduction for clarity, with the suggested passage now incorporated.

[Reviewer]: The use of the term averaging kernel (AK) is inconsistent and in some cases does not follow standard terminology.

[Response]: We have clarified our use of the AK term upon first mention (Line 255) and paid close attention to consistency through the paper.

[Reviewer]: The curves in Figure 2 [now Figure 3] do not represent the AK profiles as stated, but rather the profiles of the diagonal values of the averaging kernel matrix (AKM). More accurately, the rows of this matrix would be called averaging kernels, as it is done, e.g., in Figure 4 of Smith & Barnet (2020). There, the profiles of the diagonal values in Figure 5 are referred to as "averaging kernel diagonal vectors."

[Response]: We have updated the caption for Figure 3 (previously Figure 2) on Lines 458–462 as follows: "A statistical summary of CLIMCAPS signal-to-noise ratio (SNR) for all the temperature (left), $O_3$ (middle) and $HNO_3$ (right) retrievals north of 40˚N latitude on 2 February 2020. The profiles represent the average of the respective averaging kernel matrix diagonal vectors (AKD) with standard deviation error bars to indicate the degree to which CLIMCAPS SNR varies across retrieval pressure layers within the study region. CLIMCAPS retrieves temperature on 31 pressure levels, $O_3$ on 11 layers, and $HNO_3$ on 8 layers, hence the difference in the number of error bars across the three variables."

[Reviewer]: Figure 2 [now Figure 3]: Instead of the profiles of the diagonal values of the AKM, it would be desirable and instructive to see the AKs (i.e., the rows of the matrix). By their shapes, one can get an impression of the altitude range which contributes to the HNO3 profile at a given pressure level.

[Response]: We have added a new figure (Figure 2) on Line 284 to depict the individual AKDs for each of the experimental configurations.

[Reviewer]: p.7ff, l.202ff Please do not present the mathematical relations between the quantities in the running text, but rather use equations with numbering. This also facilitates references in the text.

[Response]: We have changed the presentation of equations accordingly.

[Reviewer]: A more detailed description of the retrieval quantities and their mathematical relations, or at least references to where the respective relations are documented, would be beneficial. Perhaps a brief recapitulation of the retrieval approach and the related quantities in a separate section earlier in the text would be appropriate.

[Response]: We have added Equation 1 as overview of the generalized Bayesian signal inversion equation on Line 188 that we then discuss in the following paragraph.

[Reviewer]: I have the most problems with the presentation of the comparison between VLIMCAPS-x and MLS HNO3 in Figure 4. Only for November and February do the spatial patterns seem to match well, while the same cannot be said for the other three months. There, the spatial patterns are clearly different. Based solely on this initial comparison, one would conclude that the desired/attempted reproduction of spatial gradients (see p.17, l.446) is not working reliably. A critical discussion of this comparison is virtually completely missing, and therefore has to be supplied in the revised manuscript. In particular, the question arises as to how future data users can recognize when the data adequately reflects stratospheric conditions and when it does not.

[Response]: We have added the following text to the discussion of Figure 5 (previously Figure 4) on Lines 579–589: "Note how the spatial patterns of CLIMCAPS $HNO_3$ strongly align with those from MLS at the onset of the vortex in November, and again as $HNO_3$ reaches its first distinct seasonal feature in February. By March, however, the CLIMCAPS $HNO_3$ feature weakens relative to that from MLS and by April is largely absent as the temperatures in the vortex start to rise. By May, the vortex has dissipated (as seen in the CLIMCAPS temperature and $O_3$ maps), along with the distinct seasonal $HNO_3$ feature in both products. It is worth noting that CLIMCAPS $HNO_3$ registers a strong low-$HNO_3$ feature (such as that visible on 2 February 2020) only when coincident with wintertime minima in both temperature and $O_3$, never outside of the conditions indicating the presence of the winter polar vortex (not shown). Additionally, note the strong agreement in spatial patterning between MLS $HNO_3$ and CLIMCAPS $O_3$ throughout the season, while the same cannot be said for the colocated CLIMCAPS HNO at this stage. It is worth reminding the reader here that a mature, optimal CLIMCAPS $HNO_3$ product does not exist yet. Figure 5 presents CLIMCAPS retrievals produced by the experimental R4 configuration, which is only a first step towards achieving a viable stratospheric nadir IR $HNO_3$ product in future. It will be interesting to determine the degree to which the CLIMCAPS $HNO_3$ retrieval can be optimized for a better correlation with MLS $HNO_3$ throughout the lifetime of the Arctic vortex."

[Reviewer]: There are the following comments regarding the technical implementation of the comparison shown in Figure 4 [now Figure 5]: First of all, it is not clear whether and how the MLS data belong to one specific fixed level or whether they have been averaged over several levels. In general, for the eye it is difficult to compare a map of values, such as those used for CLIMCAPS-x data, with the sampled MLS data. A purely visual comparison is easier to do when the CLIMCAPS-x HNO3 data is sampled, and shown, at the locations of the MLS measurements.

[Response]: We added text to clarify our presentation of the two stratospheric $HNO_3$ products. Lines 570–574: "CLIMCAPS $HNO_3$ has global coverage (90˚S to 90˚N) and is presented here as spatial averages on a 4˚ equal-angle grid. The MLS product is the Level 2 V005 $HNO_3$ mixing ratio (Manney, 2021) that has near-global coverage (82˚S to 82˚N) and profile retrievals spaced 1.5˚ along the orbital track with roughly 15 orbits per day. We integrated both profile products across their retrieval layers spanning 30–100 hPa for ease of comparison."

We note the Reviewer's preference here, but this paper does not aim to present a one-to-one comparison between colocated observations. Adding such a figure (and discussion) would distract from the overall message. Instead, we present the two products in their native form to contrast their differences in spatial coverage and vertical resolution. We have made this clear in the communication of the goals throughout the revised paper.

[Reviewer]: Of course, I understand that this presentation is mainly concerned with the development of the retrieval and not meant to anticipate a validation that is still to be carried out in future. On the other hand, the two HNO3 datasets are so different in terms of altitude resolution that I would strongly advise to transform the MLS data (regarded as "the truth"), using AKM and a priori of the CLIMCAPS-x retrieval, to that profile that would be seen by the nadir instrument/retrieval combination. I think there is no other reasonable way for a sound and robust comparison.

[Response]: Noted, but we wish to argue against employing this technique typically reserved for traditional validation studies. As stated above, this paper does not aim to present a validation of an optimized CLIMCAPS HNO$_3$ product with MLS HNO$_3$ as "the truth", because in this paper the CLIMCAPS HNO$_3$ product is not yet optimized. Instead, this paper presents the evidence that CrIS measurements (and those made by other nadir IR sounders such as AIRS and IASI) can be decomposed into eigenfunctions to allow the separation of their stratospheric and tropospheric HNO$_3$ signals. This has never been demonstrated before.

Moreover, we cannot convince funding agencies in the US to support the maturation and validation of a CLIMCAPS HNO$_3$ product without evidence that such a product is viable in the first place. This paper, therefore, presents the first and necessary step towards the maturation of a CLIMCAPS HNO$_3$ product that can be ready for validation and public release in future. Discussion has been revised throughout the manuscript to better emphasize this point.

**And finally, there are these specific points:**

[Reviewer]: p.1, ll.1-2 Shouldn't the title already contain, or refer to, "stratospheric"?

[Response]: We have changed the title to "Mapping seasonal stratospheric nitric acid (HNO$_3$) patterns in the extratropics with nadir-viewing infrared sounders – a retrieval perspective"

[Reviewer]: p.3 l.83 "MLS-like" I think this can be stated more general (e.g. "limb-viewing instrument")

[Response]: After some consideration, we maintain that the suggested change would not be suitable, since the operational and scientific success of MLS in general is not merely tied to the instrument's limb-viewing capability, but also to its spectral range and resolution. Here, we specifically mean an MLS-like instrument capability.

[Reviewer]: p.4 l.102 Strictly speaking, contrary to FORLI, MLS is no retrieval system.

[Response]: The wording on Line 132 has been changed to "has been widely adopted in many retrieval systems, including the one used for MLS and FORLI".

[Reviewer]: p.5 l.147 Parentheses seem to be unnecessary.

[Response]: The parentheses have been deleted.

[Reviewer]: Table 1 Please give the relation between lambdac and Bmax and their definitions before the quantities are used.

[Response]: We have added Equation 2 to Line 331 for an introduction of these quantities ahead of their presentation in Table 1.

[Reviewer]: p.7, l.192 & p.8, l.215 How can a SNR be destabilized?

[Response]: We added the following explanation to lines 305-309: "SNR can be destabilized when the noise is high relative to the signal, or when the noise fluctuates dramatically relative to the signal from scene-to-scene. Additionally, SNR can be destabilized when the noise (random and systematic) is not well-characterized and quantified such that it is wrongly interpreted as signal instead. Similarly, SNR can be destabilized when signal is wrongly interpreted as noise."

[Reviewer]: p.7, l.193 RTAERR=0 is not an underestimation? That means that destabilization cannot occur?

[Response]: On lines 312–319 we added the following text as clarification: "So, while RTAERR = 0 is technically an under estimation, it is close in magnitude to the real RTAERR and therefore not destabilizing. Historically, RTAERR was installed as an attempt to lower the weight of channels that had poor spectroscopic laboratory measurements or a large RTAERR value. In 1995, in the pre-launch AIRS era, this term was expected to be rather large — on the order of ~1° K for many channels — especially in the water band region. After the AIRS launch, the RTA fitting procedure was improved and more recent laboratory spectroscopy measurements were incorporated so

that over time, the RTAERR term was reduced to very low values (<0.01˚ K for most channels). We now simply ignore the few remaining channels that have high RTA errors so that setting RTAERR = 0 is no longer deemed an issue for stability."

[Reviewer]: p.8, l.205 I think that i is meant to be an index for the height level. Is it counted from top to bottom, as implied by the order of the pressure values in Table 1, or from bottom to top?

[Response]: We have changed the index to j and added a clarification on line 208.

[Reviewer]: p.9, l.246 Why do the DOFS represent the sum of all Rfac_i values? Is there a reference?

[Response]: We have removed this sentence to avoid confusing the reader.

[Reviewer]: p.12, l.333 This would be a good place to refer to the (newly introduced) equation number from Section 3.

[Response]: We have added references to equations 1–3 wherever appropriate throughout the revised text.

[Reviewer]: p.15, ll.421,422 mix-up of East & West?

[Response]: This error has been corrected.

[Reviewer]: p16,17 ll454-451 This would fit nicely into the introduction as part of a general motivation subsection.

[Response]: We have reworked the introduction accordingly.

[Reviewer]: In Figure 3 (now Figure 4), the individual maps would benefit from clearer visibility of the longitude and latitude grid.

[Response]: We agree with the reviewer in principle but wish to be careful not to over-complicate the figures. After careful deliberation, we decided to add four longitudinal labels to each map.

[Reviewer]: Why is Greenland so clearly visible in Figure 3 (now Figure 4) for R3-R5, both in dof and in stddev(dof)? And (roughly) the Pyrenees and the Bering Sea?

[Response]: We added the following text on Lines 590–596 for the sake of clarification: "One aspect that needs further investigation is how Earth surface conditions affect the $HNO_3$ retrieval. The 850–900 cm$^{-1}$ spectral region sensitive to $HNO_3$ (Figure 1) is colloquially known as the IR window region because it is predominantly sensitive to Earth surface conditions, and specifically to surface emissivity and skin temperature. This means that CLIMCAPS needs to accurately account for Earth surface conditions as a source of geophysical noise during each $HNO_3$ retrieval. That the $HNO_3$ retrievals are consistently elevated over some parts of Greenland (~45˚W), relative to the surrounding $HNO_3$ retrievals over ocean, throughout most of the season (Figure 5) is evidence that the **R4** configuration is not yet optimized. This indicates that we need to investigate how icy land surfaces are represented in the retrieval."

[Reviewer]: p.19, l83 "that the resulting HNO3 product generally reflects real atmospheric variations" is not supported by Figure 4: It is not generally true, but only under certain conditions (see my general comments).

[Response]: We have edited the sentence on Lines 662–664 to read as follows: "In a series of CLIMCAPS retrievals throughout the northern hemisphere winter of 2019/2020 using the **R4** configuration, we illustrated how CLIMCAPS $HNO_3$ compares against MLS $HNO_3$ and demonstrated that it reflects real stratospheric patterns under some conditions."

**Response to Reviewer 2**
**Citation**: https://doi.org/10.5194/egusphere-2025-1569-RC2

[Reviewer]: This manuscript explores information content and seasonal patterns of HNO3 in the upper Arctic atmosphere, as retrieved by several experimental CLIMCAPS versions applied to data collected from CrIS and ATMIS on the JPSS-1 satellite. While HNO3 is part of the operational CLIMCAPS retrieval, that portion of the retrieval has not been optimized as well as other trace gas species, and this paper outlines potential algorithm parameter updates for improved HNO3 retrieval. Due to the nearing end of life for the NASA Aura mission, other sources of stratospheric composition retrieval are needed and this study helps illustrate how IR sounders could partially fill this future information gap for HNO3.

Overall, this study is relevant and appropriate for AMT. The subject is timely given the status of NASA Aura.

[Response]: We thank the reviewer for this positive feedback. Demonstrating how IR sounders could partially fill the inevitable MLS gap for $HNO_3$ observations is the primary goal of this paper and a necessary step towards maturing an IR product for operational deployment.

[Reviewer]: I find the overall message of the paper unclear, and several aspects need improved explanation. I recommend major revisions, as I think the unclear aspects are crucial to the message of the paper.

[Response]: We have substantially reworked the paper for the sake of clarifying the overall message.

**Major revisions:**

[Reviewer]: The manuscript title suggests a general discussion of nadir IR sounders applied to HNO3 retrieval, and in the initial sections the description is aimed toward comparing nadir IR sounders to a microwave limb sounder (specifically, Aura-MLS). However, in the main body of the paper, most of the comparison is done between FORLI and CLIMCAPS, two retrieval systems applied to nadir IR sounders (IASI and CrIS, respectively). Thus the title and abstract do not seem to match the bulk of the paper, which I find primarily focused on CLIMCAPS HNO3. I would suggest changing the title and abstract to reflect that message.

[Response]: The reviewer correctly identifies that the aim of the paper is to present a nadir IR retrieval methodology, one that can yield a product of sufficient quality to help monitor extra-tropical stratospheric events once MLS on Aura is decommissioned. As stated in the revised abstract and introduction, we contrast the CLIMCAPS methodology to the more traditional approach adopted in FORLI to highlight the innovative aspect of our work. It is important to understand that the FORLI HNO3 product has not been widely adopted in the scientific community yet, and to our knowledge no evidence exists that it is able to detect northern hemispheric HNO3 associated with the winter polar vortices. Our goal, therefore, is not to present a comparison between the FORLI and CLIMCAPS products, but rather just highlight how the CLIMCAPS methodology deviates from the one adopted in FORLI to allow the separation of the stratospheric and tropospheric HNO3 IR signals. Discussion has been revised throughout the manuscript to better emphasize this point.

[Reviewer]: In much of the descriptions contrasting the FORLI and CLIMCAPS approach, it is stated that the FORLI system only retrieves a total column $HNO_3$. However, the Ronsmans 2016 and 2018 clearly state that FORLI retrieves a profile for $HNO_3$. It was simply a choice made in Ronsmans 2018 to post process the FORLI data to total columns for the analysis done in that paper. Thus, it seems misleading to characterize FORLI as is done in the manuscript; other researchers could simply choose to post process FORLI to extract partial columns in the stratosphere, which would be similar to the CLIMCAPS approach proposed here.

[Response]: We strongly disagree with the reviewer's perspective here as discussed below.

Ronsmans et al. (2016; https://doi.org/10.5194/amt-9-4783-2016) present the FORLI $HNO_3$ averaging kernels (AKs) in their paper's Figure 2. We wish to draw the reviewer's attention to the shape and pressure range of each AK function, which indicate the available IASI information content for $HNO_3$ ranging in altitude between 5 and 35 km. This means that the information retrieved at lower stratospheric layers will be correlated with information retrieved at mid-tropospheric layers. For this reason, the only reasonable "partial column" product they can calculate and validate against other products (as demonstrated later in their paper) is one that spans the same range as the AK functions, namely troposphere+stratosphere. In their Section 3, Ronsmans et al. (2016) state that the total error for

the partial FORLI HNO$_3$ column product is higher in the tropics than at mid- and polar latitudes "due to the higher concentrations of water vapor" at lower latitudes. Given that IASI, AIRS and CrIS all lack sensitivity to stratospheric water vapor, this can be interpreted as the error in tropospheric water vapor affecting the quality of the partial column HNO$_3$ product. Even if the partial column is calculated over fewer stratospheric layers, this correlated error will still affect the product because of the vertically broad AK functions. Ronsmans et al. (2018) builds on this earlier work by demonstrating the spatio-temporal patterns of FORLI HNO$_3$ over nine years. In their paper's Section 2, Ronsmans et al. (2018) state explicitly that "[b]ecause of this lack of vertical sensitivity, the HNO$_3$ total column is the most representative quantity for IASI measurements and is exploited here for the investigation of HNO$_3$ time evolution."

Our paper challenges the Ronsmans et al. (2016, 2018) conclusions by demonstrating that a correlated tropospheric+stratospheric HNO$_3$ retrieval is not inevitable — the retrieval method can be set up such that the averaging kernels peak across only stratospheric pressure layers, with negligible tropospheric correlation.

The CLIMCAPS retrieval methodology allows the separation (de-correlation) of the stratospheric and tropospheric HNO$_3$ IR signals (and noise) by employing eigenfunction decomposition of the IR measurement at every retrieval scene. In this paper, we demonstrate how the stratospheric and tropospheric HNO$_3$ IR signals (and noise) separate out into distinct eigenfunctions such that HNO$_3$ can be retrieved from the signal-to-noise-ratio (SNR) present in the stratospheric eigenfunction only. Given how Bayesian Optimal Estimation retrieval systems depend on the use of a radiative transfer model at run-time, CLIMCAPS retrieves HNO$_3$ as a full profile. But, unlike FORLI, the CLIMCAPS HNO$_3$ product can be aggregated exclusively across the stratospheric layers post-retrieval due to the fact that it has been separated from the tropospheric signal (and noise) prior to retrieval. This is a major advancement in the retrieval of HNO$_3$ from nadir IR measurements that could lead to the development of a viable and useful stratospheric product.

In summary, the FORLI and CLIMCAPS HNO$_3$ profile retrievals differ significantly in their SNRs, such that CLIMCAPS HNO$_3$ can be post-processed into a stand-alone stratospheric product, whereas FORLI HNO$_3$ is only a total-column product. Ronsmans et al. (2016, 2018) state explicitly that they integrate their HNO$_3$ profile retrievals across the full vertical column (boundary layer to top of atmosphere) in order to stabilize the SNR for the sake of a viable HNO$_3$ product. This is necessary because their retrieval approach correlates the stratospheric and tropospheric HNO$_3$ signal (and noise) present in the nadir IR measurements. To simply aggregate across the stratospheric layers of the FORLI HNO$_3$ retrieval in a post-processing step would bias the product in ways that would be difficult to quantify using traditional validation methods (i.e., comparisons against a correlative dataset).

Discussion has been revised throughout the manuscript to more clearly articulate this point.

[Reviewer]: Throughout the manuscript, the dynamic regularization approach of CLIMCAPS is claimed to more effectively separate the stratospheric and tropospheric portions of the HNO3 profile. It is unclear why this is necessarily true, and it is simply stated in the manuscript without any direct analysis or evidence to demonstrate improved separation.

[Response]: The effectiveness of the CLIMCAPS approach in separating stratospheric and tropospheric signal in the HNO$_3$ retrieval is demonstrated in Sections 3 and 4. It is not by dynamic regularization alone, but rather by the orthogonal (eigenfunction) decomposition of the IR measurement and the sub-selection of a single function ahead of each HNO$_3$ retrieval that the stratospheric and tropospheric HNO$_3$ SNRs can be separated. We have proof of this by looking at the AKM as discussed in this paper. Discussion has been revised throughout the manuscript to more clearly articulate this point.

[Reviewer]: I think a more detailed comparison of FORLI and CLIMCAPS retrieval output, compared to MLS, would be needed to demonstrate that CLIMCAPS is an improvement relative to FORLI in this way. Such a comparison is argued to be "beyond scope" for the present paper (line 104) – which is fine, but without such analysis I don't think the strat/troposphere separation is demonstrated.

[Response]: Again, we strongly disagree with the reviewer's perspective here. We employ the AKDs to clearly demonstrate how the stratospheric and tropospheric HNO3 SNR can be separated ahead of retrieval. Stated differently, a retrieval averaging kernel matrix (AKM) quantifies the SNR of the regularized measurement. If the

AKM rows (also known as Averaging Kernels) of a single retrieval indicate maxima in both the stratosphere and troposphere, then it means that the retrieval SNR is correlated across the two vertical regions. If, on the other hand, the AKs show maxima in the stratosphere only (i.e., if they approximate zero in the troposphere), then we can be sure that the retrieval product contains only stratospheric measurement SNR. This is all the proof we need. We added a figure (now Figure 2) to demonstrate and discuss the relation between the AKs and AKDs. Moreover, we have revised our discussion throughout the manuscript to more clearly articulate the value of AKDs in quantifying the benefits of the CLIMCAPS retrieval approach over that of FORLI in producing a stratospheric $HNO_3$ product.

[Reviewer]: Similarly, there are some statements made about the R3, R4 configurations being the most optimal.

[Response]: We were very careful to not use the word "optimal" in discussing any of the configurations. Instead, we present R3 and R4 as viable configurations to investigate in the future development of a scientifically useful stratospheric HNO3 nadir IR product. Following the Reviewer's comment, we made sure to clarify this in the main body of the text as well throughout the manuscript.

[Reviewer]: This argument is based purely on the DOFS and AK plots.

[Response]: Correct. As stated above and also in the paper, these information content metrics provide a wealth of information that can help us determine the SNR make-up of a retrieval product.

[Reviewer]: I am not familiar with the CLIMCAPS dynamic regularization approach, but within the OE framework, there is a similar trend in the strength of regularization vs information content. Increasing the prior variance, or reducing measurement variance in OE, would have a similar effect as increasing Bmax in the CLIMCAPS system: the DFS would increase.

[Response]: In broad terms, the DFS of a retrieval product can be manipulated in any number of ways, including those highlighted by the Reviewer. Of importance is the reason why DFS change with changes in algorithm parameters. The mere fact that two OE systems can yield DFS variations with algorithm tweaks do not make their products similar. In this paper, we demonstrate how the CLIMCAPS regularization parameter, Bmax, can be chosen such that the retrieval is made from a measurement eigenfunction that predominantly quantifies stratospheric HNO3 signal (and noise) with negligible correlation to tropospheric signal (and noise). Discussion has been revised throughout the manuscript to more clearly articulate this point.

[Reviewer]: The tradeoff that occurs is that we cannot increase the DFS too far as the retrieval can become unstable, and overly sensitive to measurement noise or forward model error;

[Response]: True, but the CLIMCAPS dynamic regularization approach largely prevents such an instability when the $B_{max}$ parameter (and thus the degree to which the measurement eigenfunctions are filtered and/or damped) is optimized for a certain retrieval variable. As discussed in previous papers (Smith and Barnet, 2019, 2020, 2025) the IR measurement SNR varies significantly across space and time with temperature (and many other geophysical factors). The CLIMCAPS dynamic optimization approach allows DFS to be high when measurement SNR for the retrieval variable is high (and vice versa). This is unlike the traditional Rodgers Optimal Estimation systems where the regularization matrix is an estimation of the HNO3 uncertainty, and specifically the inverse of $\delta x_{HNO3} \delta x_{HNO3}^T$, which is difficult to know since it implicitly depends on knowledge of the truth and it ignores the instrument's sensitivity to the background geophysical state of the Earth at the time of measurement (e.g., water vapor, clouds and Earth surface). As a long-term climate product, we do not want CLIMCAPS to be predominantly sensitive to static a priori assumptions that can become outdated over time. Moreover, we demonstrate in this paper that DFS for $HNO_3$ should not be too high lest we allow the stratospheric SNR to be correlated with tropospheric SNR.

[Reviewer]: …however, that could only be detected by examining the actual retrieved profiles, or via comparison to "truth" or validation data.

[Response]: We disagree with the Reviewer's perspective here. Validation data alone cannot yield insights into the underlying causes of observed differences between two disparate products. No change was made to the text in response to this comment.

[Reviewer]: In other words, the DFS would increase but the RMSE (compared to validation data) would also increase.

[Response]: As the Reviewer rightly identifies, DFS is an information content (i.e., SNR) metric that cannot be inflated beyond what signal is available in the measurements as this would cause the retrieval to be dominated by noise and thus a large(r) RMSE. No change was made to the text in response to this comment.

[Reviewer]: In the present manuscript, there is no analysis of the quality of the retrieval in any way, all arguments seem to be made solely from the perspective of the information side (DFS). Thus, it is unclear as to why R3 or R4 are the most optimal configurations.

[Response]: Correct. We present evidence for two configurations to pursue in future work to optimize the CLIMCAPS retrieval parameters for a scientifically viable $HNO_3$ product.

As stated earlier, we were careful to not use the term "optimal" in describing the **R3** and **R4** configurations. Instead, we provide evidence for adopting **R3**/**R4** in future work of the optimization and validation of a CLIMCAPS $HNO_3$ product. We have added some discussion in association with the new Figure 2 that provides more explanation for why **R3** and **R4** look the most promising. Also, we have extended the discussion of **R4** on p21 to clarify our reasoning.

[Reviewer]: To state this another way, we could view the main aim of the R2 – R5 tests is to "tune" the Bmax parameter. As Bmax increases, the DOFS increase – but there is no demonstration of any metric that is getting worse – only that "we argue that it is preferable for HNO3 DOFS to approximate 1.0 " (Line 402), but it is unclear WHY that is preferable.

[Response]: This paper does not present a validation study of the CLIMCAPS $HNO_3$ product, but instead we communicate our reasoning for adopting the R4 configuration in future work for optimization of the CLIMCAPS HNO3 product based on evidence that of the separation of stratospheric and tropospheric $HNO_3$ SNR. We have added an extensive discussion of DOFS on p.13 for the sake of clarification.

**Minor suggestions:**

[Reviewer]: One point that stood out was the relative difference between the prior (0.1 - 1 ppb, line 174) and the retrievals (on average - often > 14 ppb, from Figure 4) - this is more than an order of magnitude. It would be interesting to show Jacobians for these two cases: is the total optical depth of HNO3 small enough, that the Jacobian has a similar shape across this entire concentration range? Similarly, does the DOFS also change drastically across that concentration range?

[Response]: The CLIMCAPS $HNO_3$ a priori employed in this study is a static climatology and generally underestimates $HNO_3$ in the extratropics. Future work on the optimization of a CLIMCAPS stratospheric $HNO_3$ product could to include sensitivity studies with respect to different a priori estimates, and that is when the suggested Jacobian plots will add value. We focus our work in this paper, however, on analyzing the retrieval SNR (using the AKDs as quantifier) to determine the measurability of stratospheric $HNO_3$ that is largely independent of tropospheric noise using IR measurements. We have added the following clarifying text.

Lines 230–234: "The AFGL $HNO_3$ profile represents a global average ranging between 1 and 0.01 ppb in the UTLS, which is very small compared to the retrieved values. Future work will focus on re-evaluating and replacing this AFGL estimate, if necessary. For the sake of demonstrating the feasibility of a nadir-IR stratospheric $HNO_3$ product in this paper, we focus our experiments on the spectral channel sets and regularization mechanism."

Lines 234-239: "Given the large dynamic range of $HNO_3$ during the polar wintertime months, one can argue that it is preferable for $x_a$ to be small so that those regions with very low $HNO_3$ can reliably be retrieved. This is all the more important for a target variable like stratospheric $HNO_3$ that is very difficult to represent with an accurate $x_a$ at each space and time retrieval footprint. Moreover, a system like CLIMCAPS, that regularizes Eq. 1 dynamically

based on the strength of the measurement information content, yields $\hat{x} \gg x_a$ only when measurement SNR is quantifiably high."

Lines 624–627: "One could argue that a larger a priori estimate for $HNO_3$ would benefit the polar $HNO_3$ retrievals, but CLIMCAPS is a multi-user, global product suite, and we would need to carefully consider the impact of such a change to the $HNO_3$ a priori estimate on the retrieval suite as a whole since CLIMCAPS retrieves its Earth system parameters in series."

Lines 637–639: "In future, and as our knowledge of product applications evolves, we can test the sensitivity of CLIMCAPS $HNO_3$ retrievals to different a priori estimates, such as zonal estimates based on MLS retrievals or the FORLI $HNO_3$ a priori estimate, which is an aggregate profile derived from chemistry transport model fields and other retrieval systems."

Lines 663–665: "Future work will investigate how CLIMCAPS $HNO_3$ can be optimized in terms of its a priori estimate and the quantification of uncertainty in background parameters, such as Earth surface temperature and emissivity, to improve the accuracy of its observation of stratospheric $HNO_3$ under a broader range of conditions"

[Reviewer]: In Table 1, the right column that lists the pressure grid seems to suggest the trapezoid functions were different in the R1, R2 experiments, versus the R3, R4, and R5. Or, maybe this listing two different ways to describe the 8 trapezoid functions. If it is the former (that there was a change in the trapezoids as part of the experimental versions), that needs more explanation. If the latter, I would suggest splitting out the pressure level information into a separate table. Also, a figure showing the trapezoids would be helpful.

[Response]: The way that Table 1 straddled a page break in the submitted manuscript probably caused the Reviewer to misread the information. We will ask the Editorial team to ensure that the table is presented intact in the final publication.

In short, the righthand column lists the pressure hinge points (boundaries) and effective pressure of each retrieval layer so that the reader can get a sense of their vertical depth. A figure of the trapezoidal functions would merely repeat this information, so we have elected not to add it.

**Some minor technical correction:**

[Reviewer]: In line 184 "... seven of the eight broad retrieval layers" - which one was omitted, and why?

[Response]: The very top layer (0.04–9.51 hPa) was omitted merely because the vertical range of the figure spans 1000–10 hPa. We have omitted this sentence from the text to avoid confusion.

[Reviewer]: Line 220: each term here is written as "\delta x_n x_n^T", is this a typo? I think it should always be "\delta x_n \delta x_n^T"?

[Response]: We have removed this equation, replaced it with Eq. 1 (Line 188) and discuss the background error term on Line 221.

[Reviewer]: Line 235: On the fact that O3 has such a large eigenvalue, I assume the CLIMCAPS retrieval uses a targeted retrieval spectral window on the O3 band (1000-1100 1/cm)? I think that must play a role in "accentuating" the eigenvalues.

[Response]: CLIMCAPS uses channel subsets for all its retrievals and the full channel lists are published in Smith and Barnet (2025). We have not tested this in absolute terms, but agree with the reviewer's assessment that targeted channel subsets help amplify the SNR for the target retrieval variable. No change was made to the text in response to this comment.

[Reviewer]: Figure 3 & 4 (now Figures 4 & 5): the color maps here seem like they might be saturated, meaning many of the data points are beyond the max value in the color bar. (particularly for upper right panel of Fig 3, and the CLIMCAPS HNO3 maps in Figure 4). Can these be replotted with the max color bar values increased?

[Response]: For ease of interpreting the qualitative comparison presented in Figures 3 and 4, we argue that a fixed color bar range for each quantity helps illuminate the differences and similarities among the various configurations. Our goal with Figure 3 is to demonstrate how the five configurations vary in their DOFS relative to each other. Allowing variation in scale according to the values presented in each figure would impede such an inter-comparison. No change was made to the text in response to this comment.

[Reviewer]: Figure 4 [now Figure 5]: Is this the R4 CLIMCAPS test version? I don't see that listed anywhere - would help to state that directly in the figure caption.

[Response]: We have updated the Figure 5 caption as follows: "Figure 5: A time series of lower stratospheric CLIMCAPS retrievals of $T_{air}$, $O_3$, and $HNO_3$ produced by the **R4** experimental configuration aggregated onto a 4˚ equal-angle grid throughout the 2019/2020 Northern Hemisphere winter season poleward of 40˚N latitude. The column on the right represents the MLS V5 Level 2 profile $HNO_3$ retrievals at 1.5˚ intervals. Both the CLIMCAPS and the MLS profile products were vertically integrated across their respective retrieval layers spanning 30–100 hPa."

[Reviewer]: What pressure range is sampled with the MLS retrievals?

[Response]: We added the following clarification on lines 569–573: "CLIMCAPS $HNO_3$ has global coverage (90˚S to 90˚N) and is presented here as spatial averages on a 4˚ equal-angle grid. The MLS product is the Level 2 V005 $HNO_3$ mixing ratio (Manney, 2021) that has near-global coverage (82˚S to 82˚N) and profile retrievals spaced 1.5˚ along the orbital track with roughly 15 orbits per day. We integrated both profile products across their retrieval layers spanning 30–100 hPa for ease of comparison."

---

## Author Response (AR2)

I recommend three minor changes for the revised manuscript

[Reviewer]: In regards to FORLI: I still contend this FORLI does retrieve an HNO3 profile, so it is wrong to call it a "a stratosphere+troposphere integrated column product". Assuming the FORLI documentation is correct (see https://acsaf.org/docs/pum/Product_User_Manual_IASI_HNO3_Apr_2022.pdf) then the data product files contain *only HNO3 profiles*. Users have to compute the integrated quantity themselves.
It is perfectly fine to argue that this profile is best utilized as an integrated quantity (as the authors do, and in Ronsmans 2018) due to the limited sensitivity and level correlation. But the authors need to correctly describe it as such. Claiming that it is an "integrated column product" is misleading as this indicates a profile-scaling type retrieval which is not how FORLI works. The incorrect phrasing is used in the abstract and in line 88-89.

[Response]: We now refer to the FORLI product as "a stratosphere + troposphere correlated profile retrieval" in the abstract and on Line 94.

[Reviewer]: The new figure 2, showing the full averaging kernel matrices for each of the algorithm configurations, does help describe the change in algorithm behavior across configuration changes. What should be added (either as an additional figure, or new section of figure 2) is the prior and retrieved HNO3 profiles and the posterior uncertainty. This new plot would be very similar to the Figure 1 of the Ronsmans 2016 AMT paper. Thus, this manuscript would have plots similar to Figures 1 and 2 of Ronsmans 2016, and this is needed for readers to fully understand the difference between the two retrieval methodologies.

[Response]: We added the following text on Lines 308–314 to clarify our depiction of HNO3 as horizontal maps and not vertical profiles.

"*In Figure 2, the AKD profiles of the **R1**–**R4** configurations indicate that the CLIMCAPS retrieved HNO₃ profile ($\hat{x}$) will approximate the climatological $x_a$ in the troposphere irrespective of $x_{true}$ because the AKD (and therefore the SNR) approximates zeros below the tropopause. It is only in the lower stratosphere where the AKD profiles have peaks $\gg 0$ that $\hat{x}$ will significantly deviate from $x_a$. For this reason, we do not depict or promote the use of the vertica profiles because they are not representative of HNO₃ throughout the tropospheric atmosphere. Instead, we depict the retrieved HNO₃ as 2-D maps of lower stratospheric integrated values. To reiterate, the goal of our retrieval system is to retrieve stratospheric HNO₃ estimates that are independent of the tropospheric SNR in the measurements.*"

[Reviewer]: The discussion explaining the choice of prior is confusing and misleading - this is following line 232 ("Future work could focus on re-evaluating this AFGL profile...".
There is an argument here that the retrieval will be biased high if the prior is larger than the true state, "because \hat{x} >= x_a by definition", which is not a true statement - term added to x_a in Equation 1 can be negative. I think what the authors mean here is that |\hat{x} - x_a| is always smaller than |x_true - x_a|, which means that \hat{x} is always biased toward the prior.

[Response]: We removed the statement "because \hat{x} >= x_a by definition" and clarified the discussion about the AFGL profile on Lines 236–249 as follows.

"*The AFGL HNO$_3$ profile represents a global average ranging between 0.01–1.0 ppb in the UTLS, which is orders of magnitude smaller than the stratospheric values retrieved for HNO$_3$ in the extratropics during wintertime. Future work could focus on re-evaluating this AFGL profile for use as HNO$_3$ $x_a$, but the solution is not a simple replacement with a different estimate. As depicted in Eq. 1, $\hat{x}$ depends on adding measurement SNR to $x_a$, which means that whenever $x_a$ is high relative to the true state of $x$, $\hat{x}$ will be biased high. Given the large dynamic range of HNO$_3$ during the polar wintertime months, we argue that it is preferable for $x_a$ to be very small so that the retrieved product depict elevated HNO$_3$ values only where the true state ($x_{true}$) is high. Stated differently: when $x_{true}$ is very small then the corresponding IR measurement SNR for HNO$_3$ is very low and $\hat{x}$ will approximate $x_a$. Conversely, when $x_{true}$ is large, the measurement SNR is large and $\hat{x}$ will have a large departure from $x_a$. . This is all the more important for a target variable like stratospheric HNO$_3$ that is very difficult to represent with an accurate $x_a$ at each space and time retrieval footprint because of the lack of real-time observations. We can have confidence in CLIMCAPS HNO$_3$ retrievals because $\hat{x}$ will be low whenever $x_{true}$ is low, and $\hat{x}$ will significantly depart from $x_a$ only when the measurement SNR is large.*"

With those minor changes, the paper can be accepted.